# HIV-1 Vpu induces neurotoxicity by promoting Caspase 3-dependent cleavage of TDP-43

Jiaxin Yang[1], Yan Li[1], Huili Li[1], Haichen Zhang[2], Haoran Guo[1], Xiangyu Zheng[2], Xiao-Fang Yu [ID][3] & Wei Wei [ID][1,4 ✉]

## Abstract

**Despite the efficacy of highly active antiretroviral therapy in controlling the incidence and mortality of AIDS, effective interventions for HIV-1-induced neurological damage and cognitive impairment remain elusive. In this study, we found that HIV-1 infection can induce proteolytic cleavage and aberrant aggregation of TAR DNA-binding protein 43 (TDP-43), a pathological protein associated with various severe neurological disorders. The HIV-1 accessory protein Vpu was found to be responsible for the cleavage of TDP-43, as ectopic expression of Vpu alone was sufficient to induce TDP-43 cleavage, whereas HIV-1 lacking Vpu failed to cleave TDP-43. Mechanistically, the cleavage of TDP-43 at Asp89 by HIV-1 relies on Vpu-mediated activation of Caspase 3, and pharmacological inhibition of Caspase 3 activity effectively suppressed the HIV-1-induced aggregation and neurotoxicity of TDP-43. Overall, these results suggest that TDP-43 is a conserved host target of HIV-1 Vpu and provide evidence for the involvement of TDP-43 dysregulation in the neural pathogenesis of HIV-1.**

**Keywords** HIV-1; Vpu; TDP-43; Caspase 3; Neurotoxicity
**Subject Categories** Microbiology, Virology & Host Pathogen Interaction;
Neuroscience; Post-translational Modifications & Proteolysis

## Introduction

Despite considerable progress in antiretroviral therapy, HIV-1 remains a persistent global health challenge, with ~38.4 million individuals worldwide living with the virus at the end of 2021. While mortality rates associated with HIV-1 have been effectively controlled, the prevalence of complications associated with this virus remains high. HIV-associated neurocognitive disorder (HAND), encompassing a wide range of cognitive, behavioral, and motor impairments, is among the most debilitating consequences of HIV-1 infection and profoundly impacts the quality of life of affected individuals (Nightingale et al, 2014; Sreeram et al, 2022).

HIV-1 can penetrate the central nervous system (CNS) during the early stages of infection, where it can persistently replicate and form a reservoir, even in patients undergoing combined antiretroviral therapy (cART) (Saylor et al, 2016). Specific proteins encoded by HIV-1, such as gp120, Tat, Vpr, and Nef, have been previously identified as neurotoxic agents that disrupt normal neuronal function and are responsible for HAND (Hategan et al, 2017; Jones et al, 2007; Masanetz and Lehmann, 2011; Patel et al, 2000; Piller et al, 1999; Toggas et al, 1994). However, a more comprehensive understanding of the pathogenic mechanisms underlying HAND is still needed.

TAR DNA-binding protein 43 (TDP-43) is a heterogeneous nuclear ribonucleoprotein consisting of 414 amino acid proteins encoded by the *TARDBP* gene. TDP-43 plays crucial roles in RNA splicing, transcription, mRNA transport, and stabilization (Baughn et al, 2023; Prasad et al, 2019). Synthesized in the cytoplasm, TDP-43 undergoes transcription-dependent shuttling between the nucleus and the cytoplasm (Ayala et al, 2008; Sawaya et al, 2021). Under normal physiological conditions, TDP-43 is predominantly localized in the nucleus, but under pathological circumstances, it translocates to the cytoplasm, where it undergoes multiple posttranslational modifications, such as hyperphosphorylation, ubiquitination, and truncation, leading to its accumulation (Chou et al, 2018; Hasegawa et al, 2008; Neumann et al, 2006). Although TDP-43 was initially identified as an HIV-1 restriction factor that inhibits viral gene expression, it has garnered increased attention in the field of neurodegenerative diseases due to its aberrant aggregation, reduced functionality, and mislocalization, which are frequently associated with various neurological disorders (Brown et al, 2022; Lu et al, 2022; Ma et al, 2022; Ou et al, 1995; Yu et al, 2021). The presence of aggregated TDP-43 proteins has been observed within the cytoplasmic inclusions of neurons affected by various neurodegenerative disorders, including amyotrophic lateral sclerosis (ALS), frontotemporal dementia (FTD), Alzheimer's disease (AD), and Parkinson's disease (PD) (Arseni et al, 2022; Josephs et al, 2014; Ling et al, 2013; Taylor et al, 2016). TDP-43 inclusions can spread in a prion-like manner between neurons, leading to the widespread distribution of aggregated TDP-43 in the brain over time (Nonaka et al, 2013; Polymenidou and Cleveland, 2011). Recently, we and others have reported that the presence of viral proteases encoded by several RNA viruses, including SARS-CoV-2 and enteroviruses, can

[1]Institute of Virology and AIDS Research, First Hospital, Jilin University, 130021 Changchun, Jilin, China. [2]Department of Neurology and Neuroscience Center, First Hospital, Jilin University, 130021 Changchun, Jilin, China. [3]Cancer Institute (Key Laboratory of Cancer Prevention and Intervention, China National Ministry of Education), The Second Affiliated Hospital, Zhejiang University School of Medicine, Hangzhou, Zhejiang, China. [4]Key Laboratory of Organ Regeneration and Transplantation of Ministry of Education, Institute of Translational Medicine, First Hospital, Jilin University, 130021 Changchun, Jilin, China. ✉E-mail: wwei6@jlu.edu.cn

facilitate the formation of pathological TDP-43 aggregates (Fung et al, 2015; Wo et al, 2021; Yang et al, 2023; Zhang et al, 2023). However, the influence and underlying mechanisms of HIV-1 on the status of TDP-43 remain unclear.

Using cell imaging to detect the status of intracellular TDP-43 proteins, we elucidated the cleavage and aggregation of TDP-43 proteins induced by HIV-1 infection mediated through the viral accessory protein Vpu. The functional inactivation of TDP-43 by Vpu not only overcame the inhibition of HIV-1 gene expression but also triggered the formation of a dominant-negative mutant form of TDP-43 via the Vpu/Caspase 3 axis, leading to the endogenous aggregation of normal TDP-43 and significant cytotoxicity in neural cells. Given the global prevalence of HIV-1 infection, the observation that HIV-1 Vpu can induce pathological TDP-43 formation enhances our comprehension of HIV-1 cytotoxicity and provides a foundation for predicting, monitoring, and managing HIV-1-associated clinical manifestations.

## Results

### HIV-1 induces the cleavage and cytoplasmic aggregation of TDP-43

To investigate the impact of HIV-1 on TDP-43 localization and expression, we first used an HIV-1 pseudovirus to infect primary human astrocytes. The fluorescence data demonstrated that endogenous TDP-43 translocated into the cytoplasm and formed inclusions within virus-infected cells (Fig. 1A,B). The cytoplasmic aggregation of TDP-43, which is a crucial pathological feature observed in various neurodegenerative diseases, such as ALS, led us to further investigate the effect of HIV-1 infection on TDP-43.

In addition to immunofluorescent experiments for the direct detection of TDP-43 status, we also utilized a well-established cell fraction assay widely used in TDP-43 proteinopathy studies to assess the solubility of TDP-43 proteins (Winton et al, 2008; Yang et al, 2023; Zhang et al, 2023). This fractionation assay combines RIPA lysis and sonication to ensure complete disruption of both the cellular and nuclear membranes during lysis, followed by high-speed centrifugation to isolate insoluble protein components as precipitates. As expected, compared with TDP-43 proteins in the control cells, which mainly accumulated in the soluble fraction, TDP-43 proteins in the HIV-1-producing cells showed dramatically increased accumulation in the RIPA-insoluble urea-soluble fraction (Fig. 1C,D), which further supported HIV-1-mediated TDP-43 aggregation. Interestingly, we detected a smaller specific band with an approximate molecular weight of 35 kDa in HIV-1-positive cells (Fig. 1C). To examine whether this band corresponded to truncated TDP-43 proteins, expression vectors for N-terminal haemagglutinin (HA)-tagged TDP-43 were cotransfected with HIV-1 constructs into HEK293T cells. The treated cells were harvested 48 h postinfection and then prepared for immunoblotting. The results showed that TDP-43 was cleaved in the presence of HIV-1 (Fig. 1E). We also confirmed that HIV-1 induced endogenous TDP-43 cleavage (Fig. 1F). Furthermore, HIV-1 infection triggered the cleavage of endogenous TDP-43 proteins and decreased their solubility (Fig. 1G–I). We subsequently used two additional distinct TDP-43 antibodies to further validate the cleavage and cytoplasmic translocation of the TDP-43 proteins induced by HIV-1 infection (Fig. EV1). Endogenous TDP-43 cleavage was also detected in HIV-1-infected

SH-SY5Y (neural cell line), THP-1 (monocytic cell line) and Jurkat (T-cell line) cells (Appendix Fig. S1A–C). Substantial alterations in the solubility of TDP-43 proteins induced by HIV-1 infection were observed in both immortalized and primary cultured neural cells (Appendix Fig. S1D–F). We also observed variations in the solubility of HIV-1 Gag p55 among the tested cells; however, these differences did not impact the cleavage and aggregation of TDP-43 proteins induced by HIV-1 (Fig. 1C,G; Appendix Fig. S1D–F). Given the substantial impact of HIV-1 infection on the nuclear–cytoplasmic ratio of TDP-43, we conducted nuclear/cytoplasmic fractionation to demonstrate that significant aggregation of TDP-43 occurs not only in the cytoplasm but also in the nucleus in the presence of HIV-1 compared with that in uninfected cells, providing additional support for our previous experimental findings (Appendix Fig. S2). In the control cells, we observed cytoplasmic accumulation of TDP-43 in the soluble fraction due to its nuclear-cytoplasmic shuttling properties. Therefore, these results indicate that HIV-1 potently induces the formation of pathological TDP-43.

### HIV-1 Vpu is responsible for the cleavage and cytoplasmic aggregation of TDP-43

To elucidate the underlying mechanisms responsible for HIV-1-induced pathological TDP-43 formation, we conducted an unbiased screen to investigate the impact of HIV-1-encoded accessory proteins on the expression of TDP-43. Our results indicated that HIV-1 Vpu induced HA-tagged TDP-43 cleavage, whereas other viral proteins had no effect on TDP-43 expression (Fig. 2A). Consistent with this result, the introduction of an early stop codon in the Vpu sequence within a mutated HIV-1 construct resulted in failure to induce TDP-43 cleavage, unlike wild-type HIV-1 or other indicated mutated HIV-1 constructs (Fig. 2B,C).

Next, we determined the effects of viral proteins on the solubility of TDP-43 (Fig. 2D–H). Our data demonstrated that only the ectopic expression of Vpu, but not that of Vif, Vpr, or Nef, was sufficient to induce TDP-43 aggregation (Fig. 2H). The Vpu proteins from the HIV-1 groups O (MVP5180), N (YBF30), and SIVtan strains maintained similar capacities to induce the cleavage and aggregation of TDP-43 (Fig. EV2). In addition, the immunofluorescence data further suggested that Vpu expression induced TDP-43 translocation into the cytoplasm (Fig. 2I,J). Consistent with these results, compared with wild-type viruses, HIV-1ΔVpu-mutated viruses with deficient Vpu expression lost the ability to induce cytoplasmic aggregation of TDP-43 (Fig. EV3A), despite the considerably lower levels of Vpu protein expression during HIV-1 infection than in those ectopically expressing Vpu (Fig. EV3B). Therefore, these results indicate that TDP-43 is a conserved host factor that is targeted by HIV-1 Vpu.

### Vpu-triggered TDP-43 cleavage depends on Caspase 3 activation

We observed an absence of colocalization between the Vpu and cytoplasmic puncta of TDP-43 in our immunofluorescence data, suggesting that Vpu may indirectly modulate the properties of the TDP-43 protein (Appendix Fig. S3). Although little is known about Vpu-induced host protein cleavage, the cleavage and pathological aggregation of TDP-43 in neurodegenerative diseases are attributed mainly to caspase activation (Rohn, 2008; Yin et al, 2019; Zhang et al,

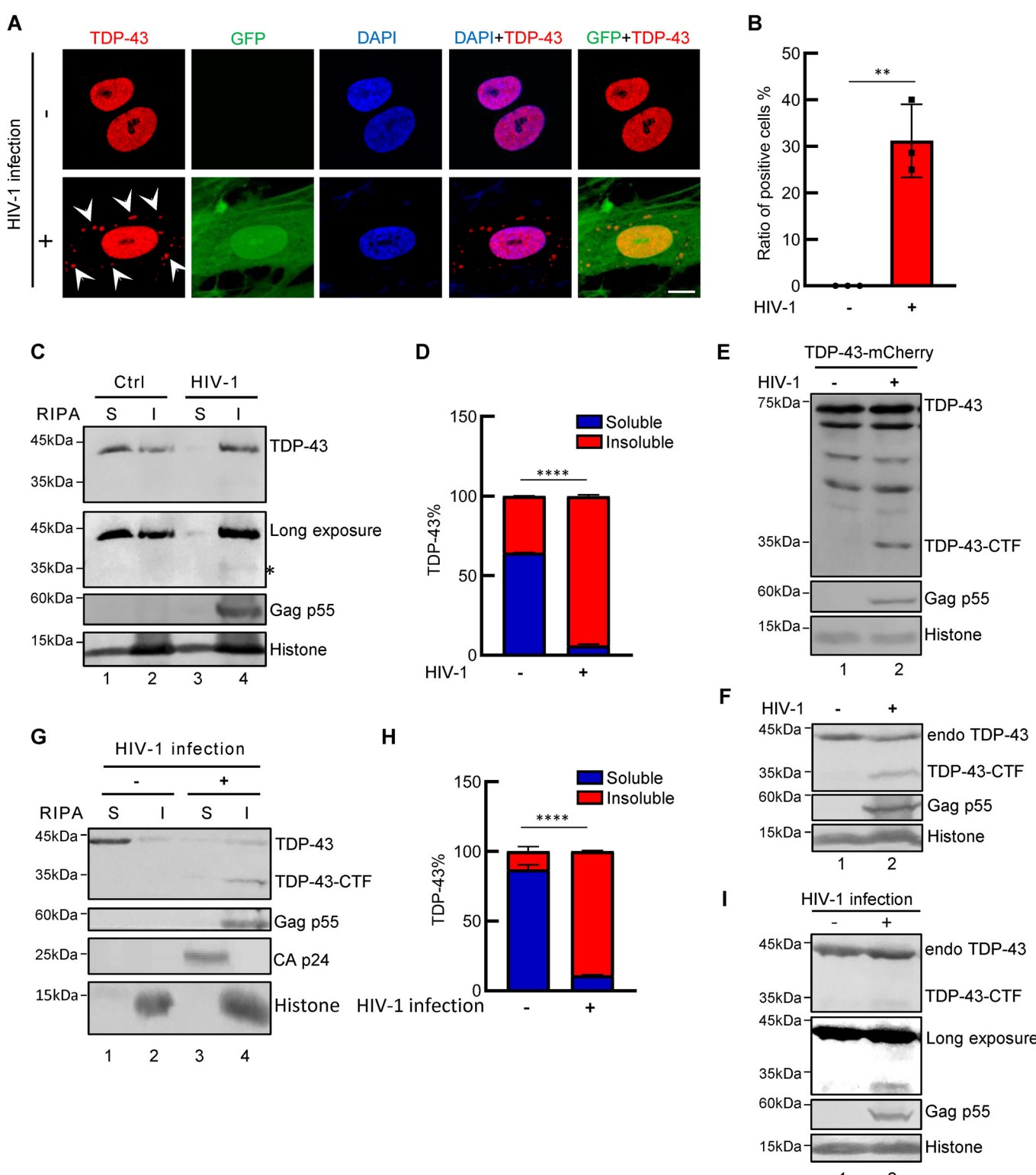

2007), which inspired us to explore the potential pathways involved in HIV-1 Vpu-mediated TDP-43 cleavage. By using different small-molecule inhibitors targeting the proteasome (MG132), lysosome (CQ) and caspases (Z-VAD), we found that the pan-Caspase inhibitor

Z-VAD efficiently blocked Vpu-mediated TDP-43 cleavage (Fig. 3A). Vpu-mediated cleavage of endogenous TDP-43 was also inhibited by Z-VAD treatment (Fig. 3B). Since Vpu stimulated endogenous Caspase 3 activation, we further confirmed that the Caspase

◀

**Figure 1.  HIV-1 induces the cleavage and cytoplasmic aggregation of TDP-43.**

(A, B) Immunofluorescence images of primary human astrocyte cells infected with the HIV-1-ΔEnv-EGFP-VSV-G virus. Red indicates endogenous TDP-43, green indicates HIV-1, and blue indicates DAPI; scale bar, 10 μm. The white arrows indicate the puncta of TDP-43. The percentages of TDP-43-positive cells in the cytoplasm are shown in the bar graph, **$P = 0.0023$. (C, D) Cell fractionation analysis of HEK293T cells transfected with the HIV-1-ΔEnv-EGFP plasmid. 48 h after transfection, proteins were sequentially extracted with RIPA buffer and 7 M urea buffer. Western blotting was performed using a TDP-43 antibody and CA p24 antibody. S soluble, I insoluble. The proportions of RIPA buffer-soluble and insoluble fractions from (C) are shown in the bar graph according to grey analysis. ImageJ was used for grey analysis, ****$P < 0.0001$. (E) Western blotting of HEK293T cells cotransfected with TDP-43-mCherry and the HIV-1-ΔEnv-EGFP plasmid. A TDP-43 antibody was used to detect TDP-43 and TDP-43-CTF. CTF C-terminal fragment. (F) Western blotting of HEK293T cells transfected with the HIV-1-ΔEnv-EGFP plasmid to detect endogenous TDP-43 using an anti-TDP-43 antibody. (G, H) Cell fractionation analysis of HEK293T cells infected with the HIV-1-ΔEnv-EGFP-VSV-G virus. ****$P < 0.0001$. (I) Western blotting of HEK293T cells infected with the HIV-1-ΔEnv-EGFP-VSV-G virus. The asterisk indicates the cleavage product of TDP-43. Data information: data are presented as mean ± SEM. ANOVA, $n = 3$ biological replicates. Source data are available online for this figure.

3-specific inhibitor Z-DEVD-FMK dramatically suppressed the Vpu-mediated cleavage of overexpressed TDP-43 and endogenous TDP-43 proteins (Fig. 3C,D). We observed that Vpu overexpression hindered the ectopic expression of TDP-43 proteins but had no effect on endogenous TDP-43 levels. This inhibitory effect of Vpu was found to be independent of Caspase 3 activation, leading us to hypothesize that Vpu may impede the functionality of elements within the expression vector. We also found that Vpu-induced TDP-43 accumulation in the RIPA-insoluble fraction was robustly reversed by treatment with Z-VAD or Z-DEVD (Fig. 3E,F). Immunofluorescence assays also confirmed that Z-VAD treatment inhibited the HIV-1-induced cytoplasmic aggregation of TDP-43 (Fig. 3G). These results indicate that Vpu-induced Caspase 3 activation is essential for HIV-1 Vpu-triggered TDP-43 cleavage and aggregation. In light of the fact that TNFα can induce Caspase 3 activation, we subjected neuronal cells to treatment with TNFα and cycloheximide (CHX). Consistently, posttreatment, we observed TDP-43 protein cleavage, a marked reduction in protein solubility, and cytoplasmic translocation in neuronal cells (Appendix Fig. S4).

## The Vpu/Caspase 3 axis induces TDP-43 cleavage at residue D89

We next explored the sites in TDP-43 proteins cleaved by Vpu and performed a sequence logo analysis of the known cleavage sites of human Caspase 3(Lavrik et al, 2005) (Fig. 4A). The results revealed two similar motifs in the TDP-43 sequences (Fig. 4B). To investigate the importance of each individual motif in the sensitivity of TDP-43 to the Vpu/Caspase 3 axis, we generated single-site mutants in the region of TDP-43 (D89E and D219E). Expression vectors for wild-type TDP-43 or the indicated mutants were cotransfected with Vpu expression plasmids into HEK293T cells. Immunoblotting assays were conducted at 48 h after transfection, and the results indicated that D89E dramatically impaired HIV-1 Vpu-mediated cleavage, whereas D219E did not influence sensitivity to Vpu compared with the expression of wild-type TDP-43 (Fig. 4C). Our observations were further supported by the finding that TDP-43-D89E was resistant to HIV-1 infection-triggered cleavage, and we also investigated HIV-1ΔVpu infection as a negative control (Fig. 4D,E). Therefore, these results indicate that the Vpu/Caspase 3 axis cleaves TDP-43 at D89.

## The Vpu/Caspase 3 axis generates a dominant-negative mutant form of TDP-43

As expected, we found that Vpu-induced TDP-43 cleavage abolished TDP-43-mediated inhibition of HIV-1 gene expression,

providing additional evidence for the arms-race relationship between viral proteins and host antiviral factors (Fig. EV4). Here, we focused on the potential impact of TDP-43 cleavage via the Vpu/Caspase 3 axis on TDP-43-mediated cellular fate. Vpu-generated TDP-43 truncation proteins (TDP-43-CTF: amino acids 90–414) were confirmed to be translocated into the cytoplasm, where they formed distinct puncta that were significantly more pronounced than those formed with the wild-type protein (Fig. 5A).

However, our repeated experiments suggested that even when the ratio of HIV-1-mediated TDP-43 cleavage was less than 10% (Fig. 5B), the solubility of endogenous full-length TDP-43 dramatically decreased in the presence of HIV-1 (Fig. 5C). We thus hypothesized that virus-generated truncated TDP-43 may act as a dominant-negative mutant. Indeed, the ectopic expression of the TDP-43 C-terminal fragment (CTF), but not wild-type TDP-43, triggered the cytoplasmic translocation of full-length TDP-43-Cherry fusion proteins (Fig. 5D,E).

Moreover, we also found that TDP-43-CTF induced the aggregation of endogenous TDP-43 (Fig. 5F,G). As a result, HIV-1 infection induced the aggregation of endogenous full-length TDP-43, which was blocked by a pan-Caspase inhibitor or specific Capase 3 inhibitor (Fig. 5H,I), suggesting that Vpu/Caspase 3 axis-dependent TDP-43 cleavage primarily causes TDP-43 aggregation by generating a dominant-negative TDP-43-truncated protein.

## Cleaved TDP-43 fragments exert cytotoxic effects on neural cells

It is well-documented that TDP-43 aggregation is associated with cell death (Gasset-Rosa et al, 2019; Zhang et al, 2009). Using the release of lactate dehydrogenase (LDH) from cells as an indicator of necrotic cell death (Zhang et al, 2009), we found that Vpu-generated TDP-43 fragments were much more toxic than wild-type TDP-43 in T98G glioblastoma cells (Fig. 6A). In addition, the expression of truncated TDP-43 fragments obviously disrupted the normal proliferation and morphology of neural cells (Fig. 6B).

These results led us to examine the cytotoxicity of HIV-1 Vpu. Our data showed that Vpu itself was capable of exerting cytotoxicity to cells, while the loss of the Vpu gene weakened HIV-1 infection-mediated neurotoxicity in T98G glioblastoma cells and SH-SY5Y neuroblastoma cells (Fig. 6C,D). In addition, downregulating TDP-43 relieved Vpu cytotoxicity (Appendix Fig. S5). As Vpu-mediated TDP-43 cleavage relies on Caspase 3 activity, we further confirmed that the small-molecule inhibitor Z-DEVD significantly relieved Vpu cyto-toxicity (Fig. 6E,F). Importantly, we observed that HIV-1 Vpu also

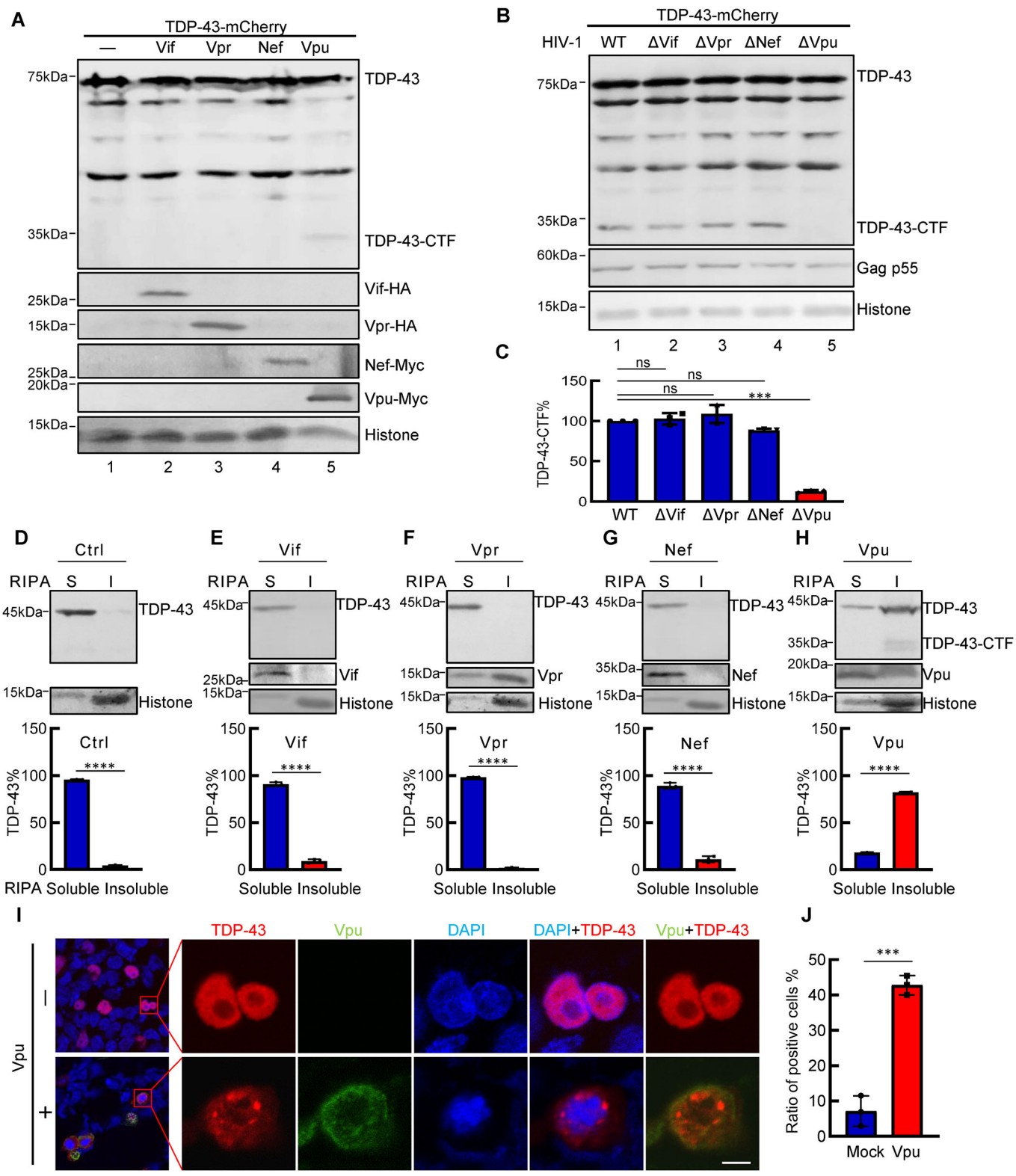

induces the cleavage of Golgi-localized, gamma ear-containing, ARF-binding protein 3 (GGA3), which is closely associated with neurodegeneration due to its crucial role in the degradation of BACE and reduction in Aβ peptide production (Fig. EV5). We then further

confirmed that Vpu-induced GGA3 cleavage is dependent on Caspase activity (Fig. EV5C,D). These results indicate that Vpu is another neurotoxic factor of HIV-1 that plays a crucial role in inducing neurotoxicity.

**Figure 2. Viral proteins involved in HIV-1-induced cleavage and aggregation of TDP-43.**

(A) Western blotting of HEK293T cells cotransfected with TDP-43-mCherry and expression plasmids for the HIV-1 accessory proteins Vif/Vpr/Nef/Vpu. The antibodies used are shown. (B, C) Western blotting of HEK293T cells cotransfected with TDP-43-mCherry and plasmids for the expression of the wild-type HIV-1 proviral gene or a deletion mutant. The bar graph shows the percentages of the relative band intensities for the CTF of TDP-43 according to grey analysis. The CTF of TDP-43 cotransfected with wild-type HIV-1 was set to 100%, ***$P = 0.0001$, ns not significant. (D–H) Cell fractionation analysis of HEK293T cells cotransfected with empty vector (D) or expression plasmids for Vif (E), Vpr (F), Nef (G), and Vpu (H). Western blotting was performed using the antibodies shown. The proportions of RIPA buffer-soluble and insoluble fractions among total TDP-43 are shown in the bar graph according to grey analysis. S soluble, I insoluble. ****$P < 0.0001$. (I, J) Immunofluorescence images of HEK293T cells cotransfected with the pmC1-TDP-43 and VR1012-HIV-1-Vpu plasmids. Red indicates TDP-43, green indicates HIV-1 Vpu, and blue indicates DAPI; scale bar, 5 μm. The percentages of TDP-43-positive cells in the cytoplasm are shown in the bar graph, ***$P = 0.0003$. Data information: data are presented as mean ± SEM. ANOVA, $n = 3$ biological replicates. Source data are available online for this figure.

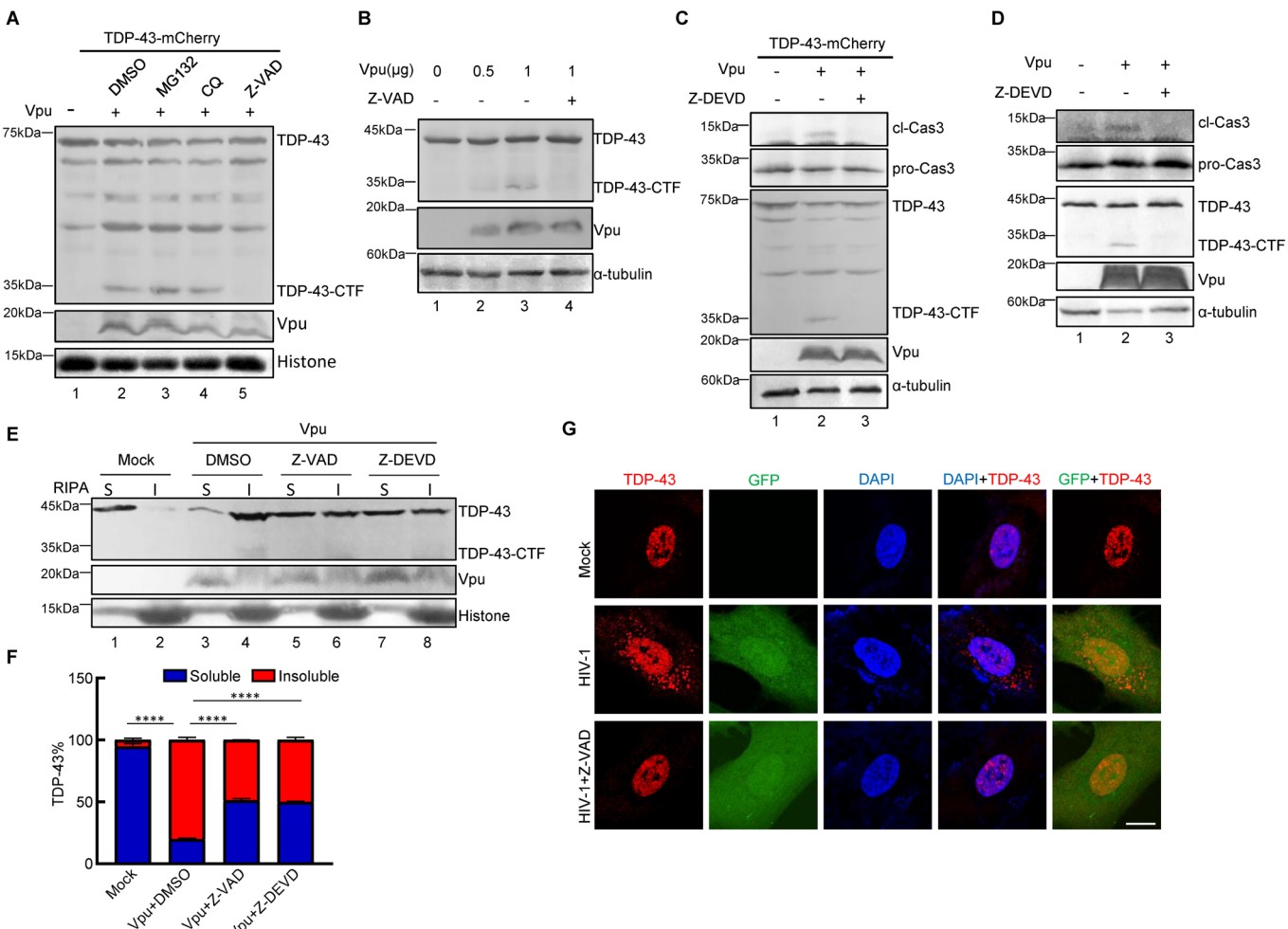

**Figure 3. Caspase 3 activation is essential for HIV-1 Vpu-triggered TDP-43 cleavage and aggregation.**

(A) Influence of inhibitors on the cleavage of TDP-43 induced by Vpu. HEK293T cells were cotransfected with TDP-43-mCherry and VR1012-Vpu-Myc and then treated with the inhibitors marked in the figure. (B) Western blotting of HEK293T cells transfected with increasing doses of VR1012-Vpu-Myc and then treated with 20 μM Z-VAD-FMK at 4 h post transfection. (C, D) Western blotting of HEK293T cells transfected with VR1012-Vpu-Myc and pmC1-TDP-43-HA (C) or VR1012-Vpu-Myc alone (D) and treated with DMSO or 20 μM Z-DEVD-FMK at 4 h posttransfection. A Caspase 3 antibody was used to detect pro-Caspase 3 and cl-Caspase 3. Cas3 Caspase 3, cl cleaved. (E, F) Cell fractionation analysis of HEK293T cells transfected with VR1012-Vpu-Myc and then treated with DMSO, Z-VAD-FMK (20 μM) or Z-DEVD-FMK (20 μM). The proportions of RIPA buffer-soluble and insoluble fractions are shown in the bar graph according to grey analysis. S soluble, I insoluble. ****$P < 0.0001$. (G) Immunofluorescence images of primary human astrocyte cells infected with the HIV-1-ΔEnv-EGFP-VSV-G virus. The cells were treated with DMSO or Z-VAD-FMK (20 μM) at 24 h post infection. Red indicates endogenous TDP-43, green indicates HIV-1, and blue indicates DAPI; scale bar, 10 μm. Data information: data are presented as mean ± SEM. ANOVA, $n = 3$ biological replicates. Source data are available online for this figure.

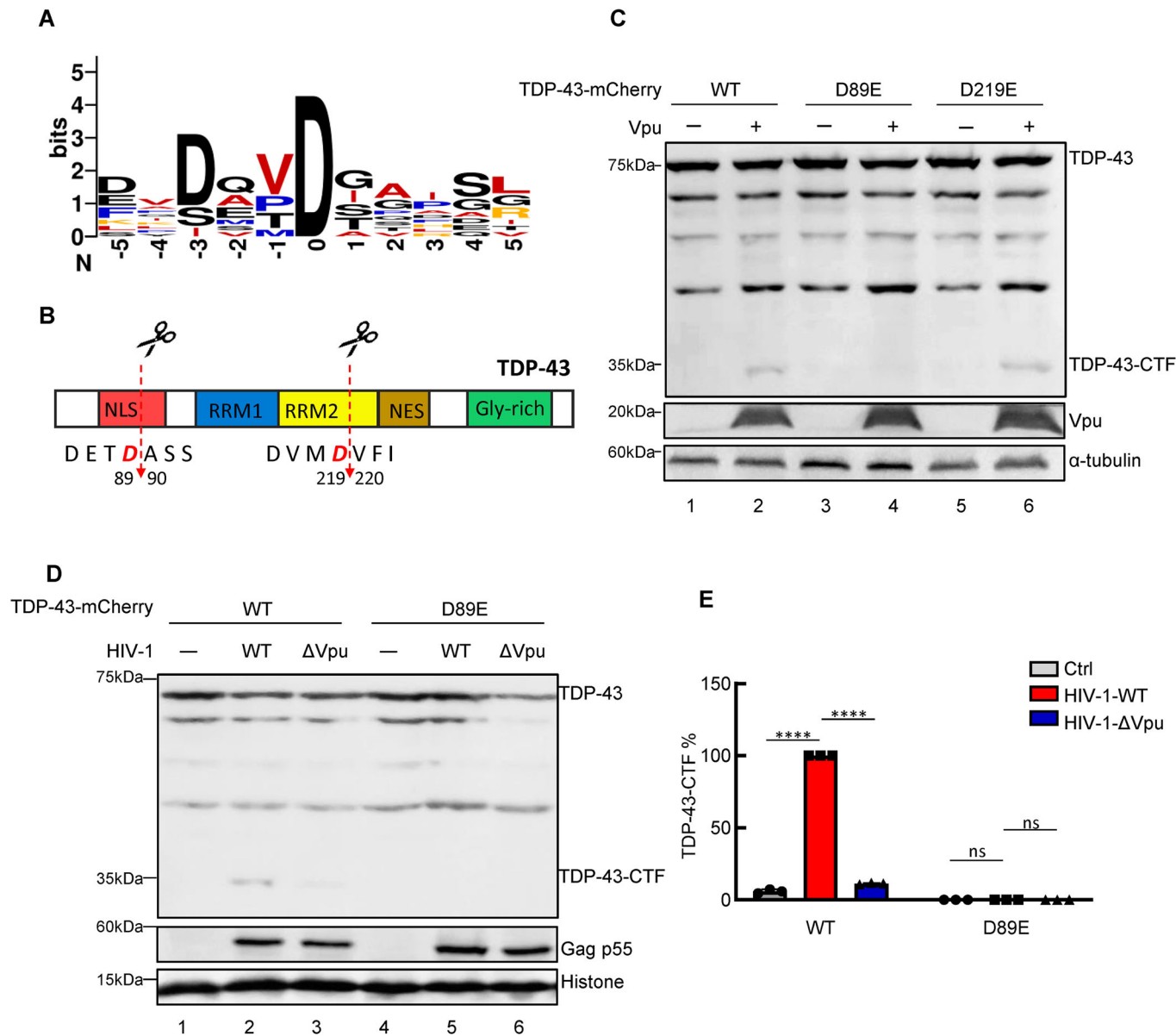

**Figure 4. Identification of TDP-43 cleavage sites utilized by HIV-1 Vpu.**

(A) Logo analysis of the cleavage site utilized by Caspase 3 by WebLogo. (B) Schematic of the domains and motifs of TDP-43 depicting the two Caspase 3 recognition motifs. NLS nuclear localization signal, NES nuclear export signal, RRM1 RNA recognition motif 1, RRM2 RNA recognition motif 2, Gly-rich glycine-rich domain. (C) Western blotting of HEK293T cells cotransfected with VR1012-Vpu-Myc and wild-type or mutant pmC1-TDP-43. The antibodies used are shown. (D, E) Western blotting of HEK293T cells cotransfected with HIV-1 wild-type or ΔVpu and wild-type or D89E mutant pmC1-TDP-43. The bar graph shows the percentages of the relative band intensities for the CTF of TDP-43 according to grey analysis. The cleavage product of TDP-43 triggered by wild-type HIV-1 was set to 100%. ImageJ was used for grey analysis. ****P < 0.0001, ns not significant. Data information: data are presented as mean ± SEM. ANOVA, n = 3 biological replicates. Source data are available online for this figure.

## Discussion

The introduction of cART has transformed HIV-1 into a chronic disease with a life expectancy approaching that of the general population for patients who adhere to treatment (Saylor et al, 2016). Nevertheless, despite the use of antiretroviral medications, cross-sectional investigations suggest that ~50% of individuals afflicted with HIV-1 encounter diverse levels of cognitive dysfunction (Clifford and Ances, 2013; Nightingale et al, 2014;

Sreeram et al, 2022). A staggering 38.4 million individuals worldwide are afflicted with chronic HIV-1 infection, making the mitigation of neurological damage caused by HIV-1 infection a challenging yet imperative task. However, thus far, the etiology of HIV-associated neurological disorders remains controversial and unclear (Saylor et al, 2016). In the present study, we found that HIV-1 can trigger pathological TDP-43 formation to subsequently generate neurotoxicity via the Vpu/Caspase 3 axis (Fig. 7).

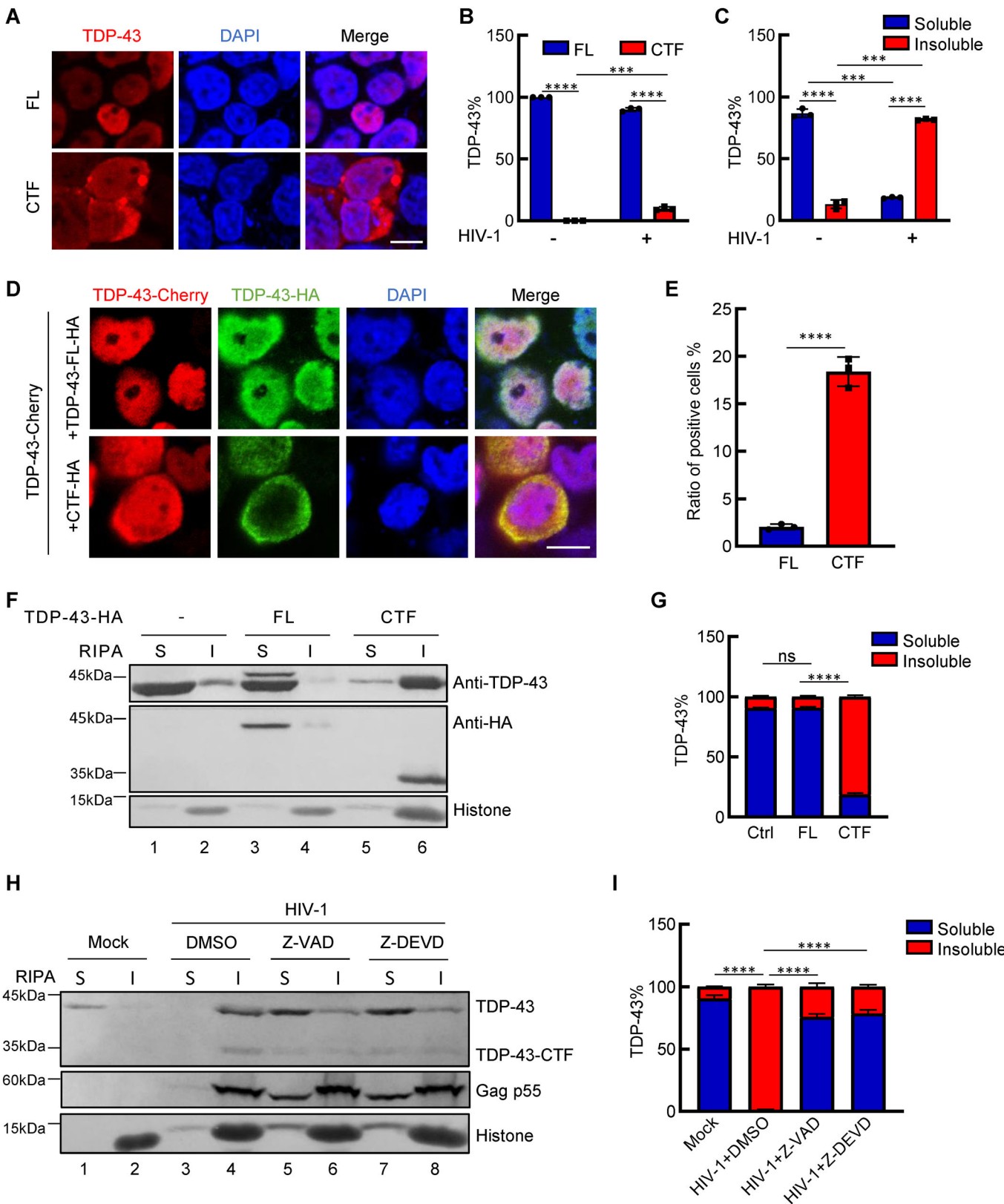

◀ **Figure 5.  The CTF of TDP-43 causes native protein aggregation.**

(A) Immunofluorescence images of TDP-43-FL-HA or TDP-43-CTF-HA in HEK293T cells. An HA antibody was used to detect TDP-43. Red indicates TDP-43, and blue indicates DAPI. (B) Analysis of the proportions of FL TDP-43 and CTF TDP-43 in cells infected with HIV-1-ΔEnv-EGFP-VSV-G. ***$P = 0.0003$, ****$P < 0.0001$
(C) Analysis of the proportions of RIPA-soluble and insoluble fractions of TDP-43 in cells after HIV-1-ΔEnv-EGFP-VSV-G infection, ***$P = 0.0001$, ****$P < 0.0001$.
(D, E) Immunofluorescence images of HEK293T cells cotransfected with pmC1-TDP-43 and VR1012-TDP-43-FL/CTF-HA. HA and Alexa Fluor 488-conjugated secondary antibodies were used to detect TDP-43-FL/CTF-HA. In the cells used to investigate total colocalization, those in which Cherry-TDP-43 was located in the cytoplasm were considered positive. The bar graph shows the ratios of positive cells. ****$P < 0.0001$. (F) Cell fractionation analysis of HEK293T cells transfected with VR1012-TDP-43-FL-HA or VR1012-TDP-43-CTF-HA. Western blotting was performed with the indicated antibodies. S soluble, I insoluble. (G) The proportions of RIPA buffer-soluble and insoluble fractions are shown in the bar graph according to grey analysis from (F), ****$P < 0.0001$, ns not significant. (H, I) Cell fractionation analysis of HEK293T cells infected with HIV-1-ΔEnv-EGFP-VSV-G and then treated with DMSO, Z-VAD-FMK (20 μM) or Z-DEVD-FMK (20 μM). Scale bar, 10 μm. ****$P < 0.0001$. Data information: data are presented as mean ± SEM. ANOVA, $n = 3$ biological replicates. Source data are available online for this figure.

Indeed, HIV-1 viruses can traverse the blood–brain barrier within the initial two weeks following primary infection by utilizing migrating myeloid and lymphoid cells to establish persistent infection in perivascular macrophages and microglia (Resnick et al, 1988; Siddiqui et al, 2022). Latent HIV-1 viruses persist in the brain even when systemic virological control is achieved with cART (Sreeram et al, 2022). Currently, the neurological damage caused by HIV-1 is attributed primarily to two factors: first, it is directly induced by neurotoxic viral proteins such as Tat and gp120 (Hategan et al, 2017; Toggas et al, 1994); second, it is indirectly triggered by neuroinflammation resulting from excessive microglial activation (Siddiqui et al, 2022). The advancement of further research is imperative to enhance our comprehension of HIV-1-associated neurological disorders. Our findings that Vpu is a neurotoxic protein provide a novel perspective for understanding the neurotoxicity of HIV-1, revealing for the first time the potential for crosstalk between HIV-1 Vpu and host TDP-43. The presence of cytoplasmic TDP-43 aggregates in neural cells from HIV-1-positive patients, as observed in previous clinical studies (Douville and Nath, 2017; Satin and Bayat, 2021; Verma and Berger, 2006), strongly supports our findings. Reciprocally, we provide a comprehensive explanation for this aforementioned clinical observation.

The initial discovery of TDP-43 revealed its role as a restriction factor that binds the HIV-1 LTR region, effectively suppressing HIV-1 transcription and gene expression (Ou et al, 1995). In response, HIV-1 employs Vpu to induce the cleavage and inactivation of TDP-43 proteins, thereby eliminating their ability to inhibit HIV-1 gene expression (Fig. EV4). The interaction between Vpu and TDP-43 provides additional evidence supporting the interplay between viral proteins and host restriction factors in the context of an arms-race relationship. The viral protein Vpu, encoded by HIV-1 and certain simian immunodeficiency viruses (SIVs) but not by HIV-2, is primarily recognized for its contribution to virion release through counteracting the host restriction factor tetherin (also known as BST-2 or CD317), which normally impedes the release of virions by physically anchoring them to the cell surface; however, Vpu antagonizes the effect of tetherin, thereby facilitating efficient release (Neil et al, 2008).

In addition, Vpu counteracts the antiviral activity of another host factor, PSGL1, and induces the degradation of the viral receptor CD4, as well as the inhibition of the activation of the transcription factor nuclear factor kappa B (Fu et al, 2020; Langer et al, 2019; Liu et al, 2019; Murakami et al, 2020; Willey et al, 1992). The present study demonstrates that TDP-43 is another conserved

target of Vpu proteins derived from various strains of HIV-1 and SIV (Fig. EV2), highlighting the novel role of Vpu in facilitating the disruption of viral gene expression barriers by HIV-1.

Recently, we and others have reported that infection with diverse human RNA viruses, including SARS-CoV-2, enterovirus D68, and coxsackieviruses, can induce the aggregation of TDP-43 in infected cells (Fung et al, 2015; Wo et al, 2021; Yang et al, 2023; Zhang et al, 2023). One feature common to these related RNA viruses is that these viruses encode their own proteases to specifically cleave TDP-43. The present study reveals the dependence of HIV-1-induced TDP-43 cleavage and aggregation on Vpu, which lacks proteolytic effects. The viral protein Vpu targets TDP-43 by indirectly enhancing Caspase 3 protease activity, indicating that a shared objective among diverse human viruses is to regulate the stability and status of TDP-43 proteins through distinct strategies. Indeed, previous functional studies on Vpu have demonstrated that HIV-1 Vpu can increase the level of activated Caspase 3 by suppressing the NF-κB-dependent expression of antiapoptotic genes, and our study further corroborates and expands upon these findings (Akari, 2001). Besides, the HIV-1 envelope can also induce the activation of Caspase 3, which potentially enhances the cleavage of TDP-43 (Cicala et al, 2000). An additional accessory protein of HIV-1, Vpr, can activate Caspase 3 (Stewart et al, 2000). However, we did not observe a significant influence of Vpr on the status of TDP-43 proteins. Further investigations are warranted to elucidate whether additional factors are involved in Vpu/Caspase 3 axis-mediated TDP-43 cleavage. Notably, in addition to the observed increase in TDP-43 inclusions in virus-infected cells, H1N1 influenza infection induces α-synuclein aggregation, which is another significant hallmark of neurodegenerative diseases (Marreiros et al, 2020). The aforementioned studies provide compelling avenues for future investigations into viral neurotoxicity, necessitating vigilant monitoring of alterations in protein stability induced by viral infection and the underlying mechanisms. Indeed, we have demonstrated that HIV-1 Vpu also induces the cleavage of GGA3 in a caspase-dependent manner (Fig. EV5). These findings underscore the importance of considering the neurotoxicity mediated by the HIV-1 Vpu/Caspase 3 axis, which extends beyond TDP-43.

Interestingly, the utilization of Caspase 3 to cleave TDP-43 proteins is not unique to HIV-1 Vpu but may instead mimic the existing interactions of host proteins (Zhang et al, 2007). The cleavage of TDP-43 observed in postmortem brains affected by AD and PD has been demonstrated to be Caspase-dependent (Kokoulina and Rohn, 2010; Rohn, 2008). A bipartite classic

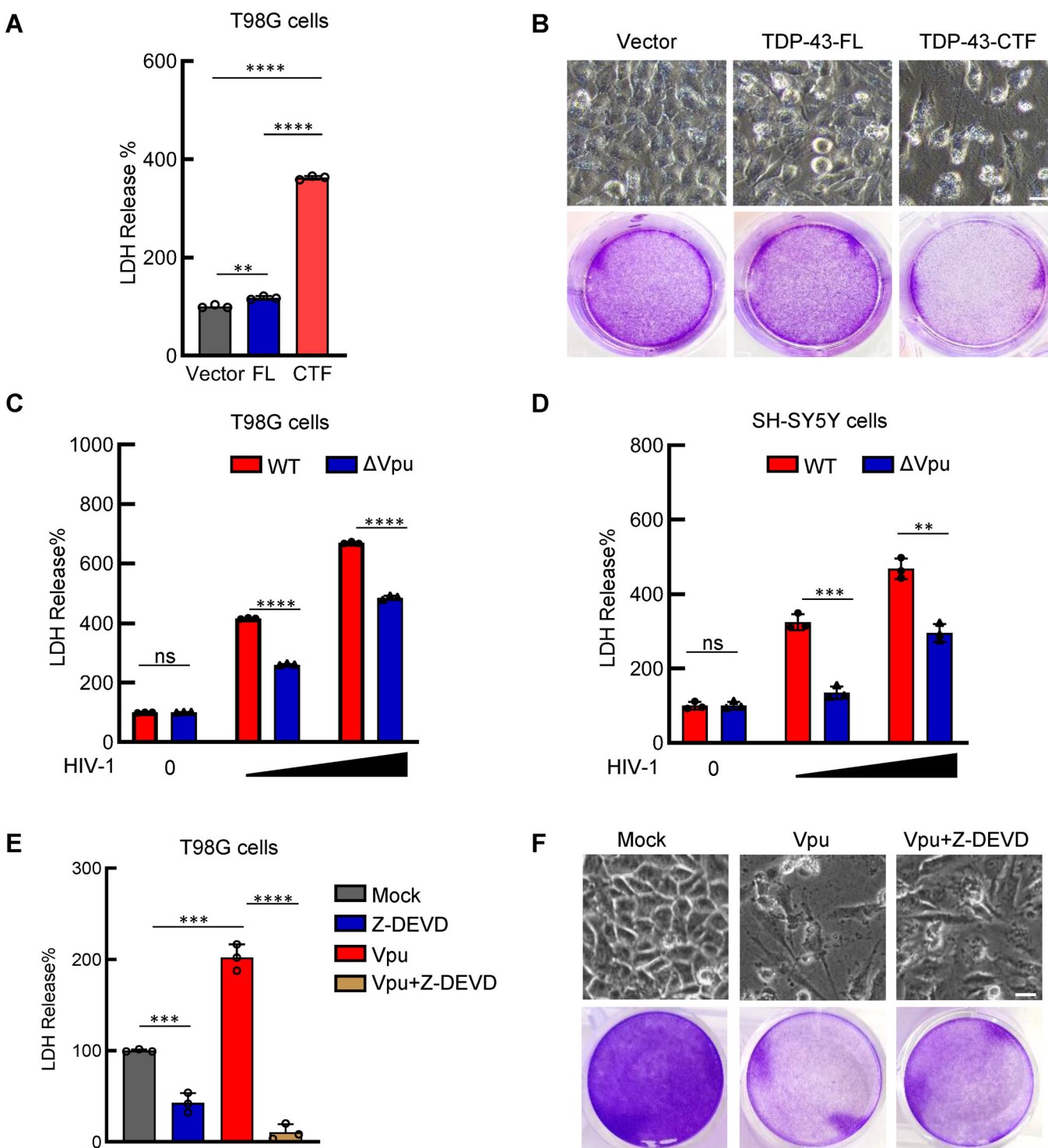

**Figure 6. Vpu causes cellular toxicity in nerve cells.**

(A) LDH release induced by FL or CTF TDP-43 in T98G cells. VR1012-TDP-43-FL/CTF-HA was transfected into T98G cells. 72 h posttransfection, the supernatants were collected to measure the level of LDH release using an LDH assay kit, **$P = 0.0031$, ****$P < 0.0001$. (B) Bright-field (upper panel) and crystal violet staining (lower panel) images of T98G cells transfected with VR1012-TDP-43-FL/CTF-HA. Scale bar, 10 μm. (C, D) LDH release from T98G (C) and SH-SY5Y (D) cells infected with wild-type or Vpu-defective HIV-1-ΔEnv-VSV-G virus, ns not significant, ****$P < 0.0001$, ***$P = 0.0003$, **$P = 0.0012$. (E, F) LDH release and images of Vpu and T98G cells treated with Z-DEVD-FMK (20 μM). Z-DEVD ***$P = 0.0008$, Vpu ***$P = 0.0003$, ****$P < 0.0001$. Data information: data are presented as mean ± SEM. ANOVA, $n = 3$ biological replicates. Source data are available online for this figure.

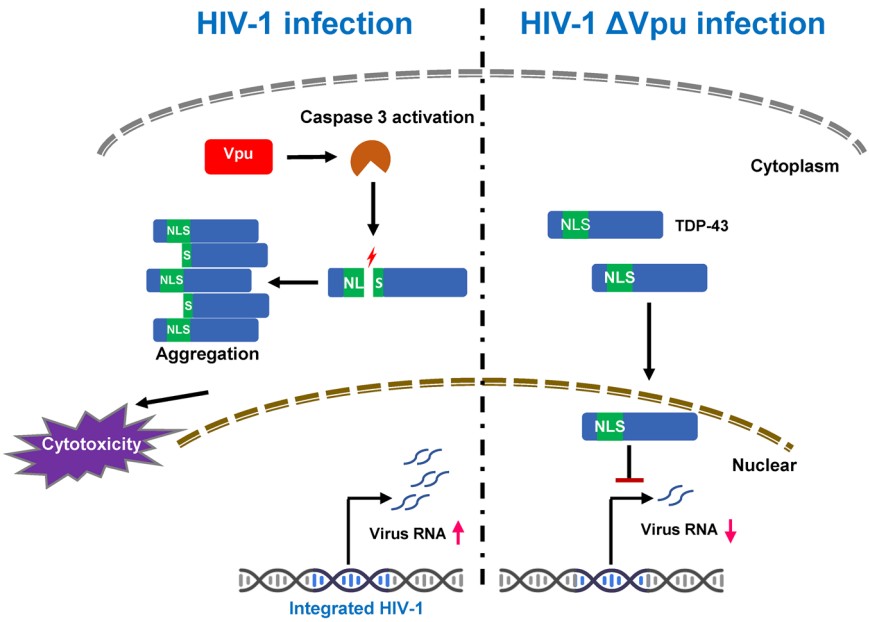

**Figure 7. Model showing how HIV-1 Vpu manipulates the TDP-43 pathway to increase transcription and release cytotoxicity.**

After HIV-1 infection of host cells, the accessory protein Vpu activates Caspase 3, and then activated Caspase 3 induces the cleavage of TDP-43 at Asp89. This process enhances the transcription of the integrated HIV-1 genome. More importantly, TDP-43 cleavage induces the aggregation of TDP-43 and neurotoxicity, which induces glial cell death.

nuclear localization signal (NLS) composed of K82RK84 (NLS1) and K95VKR98 (NLS2) has been identified at the N-terminus of TDP-43 (Ayala et al, 2008). We confirmed that the Vpu/Caspase 3 axis cleaves TDP-43 at Asp89, disrupting its NLS1 domain. Consequently, the defective TDP-43 fragments with impaired NLS1 cannot be fully translocated into the nucleus, leading to their accumulation and sequestration as cytoplasmic aggregates. Aggregated TDP-43 is unable to effectively suppress viral gene expression and exerts additional cytotoxic effects on neural cells (Figs. 6 and EV4). Moreover, we identified Vpu/Caspase 3-generated TDP-43-CTF as a potent dominant-negative mutant capable of robustly inducing normal TDP-43 aggregation, which enables HIV-1 to efficiently trigger TDP-43 aggregation despite the inefficiency of cleaving TDP-43. A major role of TDP-43 is to repress the inclusion of cryptic exons during RNA splicing. Dysfunction of TDP-43, which is an early pathological feature of TDP-43-associated ALS (ALS-TDP) and FTLD-TDP, leads to derepression of nonconserved intronic sequences that are erroneously included in mature RNAs. These events, known as cryptic exons (CEs), often result in premature stop codons and transcript degradation or premature polyadenylation. In addition, we confirmed that HIV-1 infection enhances cryptic exon inclusion in the FTD-ALS gene UNC13A and other genes, such as ATG4B and GSPM2 (Brown et al, 2022; Ling et al, 2015; Ma et al, 2022), establishing a functional connection between HIV-1 infection and loss of TDP-43 function (Appendix Fig. S6). A benefit of our study is that pharmacological inhibition of Caspase 3 activity represents a potential therapeutic strategy for alleviating Vpu-induced neural damage, and further validation through preclinical investigations is needed.

TDP-43 is associated primarily with ALS and a subtype of FTD, and several studies have noted its presence in the brains of elderly individuals affected by various neurodegenerative diseases, including AD, PD, and ALS (Meneses et al, 2021; Van Deerlin et al, 2008; Yamashita et al, 2022). Understanding the duration of virus-induced pathological TDP-43 formation is crucial for future investigations, considering that some viruses cause transient infections, whereas others persist in the human body. These investigations will contribute to the evaluation of the neurotoxic effects caused by specific viral infections. However, HIV-1 exhibits intermittent cycles of latent reactivation and silencing in the brain, continuously increasing the vulnerability of the CNS to viral disruption. Therefore, monitoring HIV-1-induced TDP-43 aggregation is imperative.

Due to the species-specific nature of HIV-1 infection, direct detection of its impact on nervous system function in existing mouse models has been challenging (Kitsera et al, 2023). However, by investigating viral factors associated with HIV-1 neurotoxicity, such as exogenously expressed Vpu, Tat, and gp120, in vivo, establishing a mouse model for neurodegenerative diseases caused by viral proteins has become an intriguing research direction. The successful establishment of relevant animal models in future studies would not only contribute to understanding viral neurotoxicity but also may advance TDP-43-related neuroscience.

Taken together, our results demonstrate that HIV-1 Vpu specifically stimulates Caspase 3 activity, following which Caspase 3 can subsequently cleave TDP-43, leading to TDP-43 aggregation and cytotoxicity in human neural cells. In light of the crucial role of pathological TDP-43 formation in neural disorders, our studies underscore the necessity of closely considering the neurotoxicity of Vpu during the treatment of HIV-1-related neuroclinical symptoms.

# Methods

## Reagents and tools table

| Reagent/resource | Reference or source | Identifier or catalog number |
|---|---|---|
| **Experimental models** | | |
| HEK293T cells (*H. sapiens*) | ATCC | Cat#CRL-3216 |
| T98G cells (*H. sapiens*) | ATCC | Cat#CRL-1690 |
| THP-1 cells (*H. sapiens*) | National Collection of Authenticated Cell Cultures | Cat#SCSP-567 |
| Jurkat cells (*H. sapiens*) | National Collection of Authenticated Cell Cultures | Cat#SCSP-513 |
| SH-SY5Y cells (*H. sapiens*) | National Collection of Authenticated Cell Cultures | Cat#SCSP-5014 |
| Primary human astrocyte cells (*H. sapiens*) | iCell Bioscience Inc | Cat#PriMed-iCell-007 |
| **Recombinant DNA** | | |
| pmC1-TDP-43 | Generay Biotech Co. Ltd. | N/A |
| VR1012-TDP-43-N-HA | Generay Biotech Co. Ltd. | N/A |
| pshRNA-TDP-43 | Generay Biotech Co. Ltd. | N/A |
| pNL4-3-ΔEnv-EGFP | NIH AIDS Research and Reference Reagent Program | Cat#111000 |
| VR1012-HA-MVP5180-Vpu | Sangon Biotech | N/A |
| VR1012-HA-YBF30-Vpu | Sangon Biotech | N/A |
| VR1012- HA-SIVcpz_TAN1-Vpu | Sangon Biotech | N/A |
| Replication-defective pseudovirus DHIV plasmids | Vicente Planelles (Bonczkowski et al, 2014) | N/A |
| pCMV-VSV-G | Addgene | Cat#8454 |
| pRSVRev | Addgene | Cat#12253 |
| pMDLg/pRRE | Addgene | Cat#12251 |
| VR1012-TDP-43-CTF-HA | This study | N/A |
| pmC1-TDP-43-D89E/D219E | This study | N/A |
| **Antibodies** | | |
| Mouse monoclonal anti-TDP-43 (WB:1/5000,IF:1/500) | Abcam | Cat#ab104223 |
| Rabbit polyclonal anti-TDP-43 (WB:1/2000,IF:1/3000) | Proteintech | Cat#10782-2-AP |
| Rabbit polyclonal anti-TDP-43 (C-terminal) (WB:1/500, IF:1/2000) | Proteintech | Cat#12892-1-AP |
| Rabbit polyclonal anti-Vpu (WB:1/1000, IF:1/250) | Abcam | Cat#ab81532 |
| Rabbit polyclonal anti-Caspase-3 (WB:1/1000) | Cell Signaling Technology | Cat#9662 |
| Mouse monoclonal anti-HA (WB:1/5000, IF:1/1000) | Covance | Cat#MMS-101R-1000 |

| Reagent/resource | Reference or source | Identifier or catalog number |
|---|---|---|
| Rabbit monoclonal anti-Myc (WB:1/1000, IF:1/100) | Abcam | Cat#ab32072 |
| Mouse monoclonal anti-alpha Tubulin (WB:1/5000) | Abcam | Cat# ab7291 |
| Rabbit monoclonal anti-GGA3 (WB:1/1000) | Abcam | Cat#ab180951 |
| Rabbit monoclonal anti-Histone H3 (WB:1/10000) | Abcam | Cat#ab176842 |
| Mouse monoclonal anti-p24 (WB:1/1000, IF:1/1000) | Abcam | Cat#ab9701 |
| Goat polyclonal anti-Rabbit IgG (H + L) Highly Cross-Adsorbed Secondary Antibody, Alexa Fluor™ Plus 488 (IF:1/1000) | Life Technologies | Cat#A-32731 |
| Goat polyclonal anti-Mouse IgG (H + L) Cross-Adsorbed Secondary Antibody, Alexa Fluor™ 594 | Life Technologies | Cat#A-11005 |
| **Oligonucleotides and other sequence-based reagents** | | |
| qPCR primers | This study | Table S1 |
| **Chemicals, enzymes and other reagents** | | |
| MG132 | MedChemExpress | Cat#HY-13259 |
| Chloroquine | MedChemExpress | Cat#HY-17589A |
| Z-VAD-FMK | MedChemExpress | Cat#HY-16658B |
| Z-DEVD-FMK | MedChemExpress | Cat#HY-12466 |
| TNFα | Abcam | Cat#ab9642 |
| Cycloheximide | SelleckChem | Cat#S7418 |
| DAPI | Sigma | Cat#F6057 |
| Complete protease inhibitor cocktail tablets | Roche | Cat#11697498001 |
| **Software** | | |
| WebLogo 3 | https://weblogo.threeplusone.com/ | N/A |
| MEGA7 | https://www.megasoftware.net/ | N/A |
| **Other** | | |
| Hieff Clone Universal One Step Cloning Kit | Yeasen Biotechnology, Shanghai, CN | Cat#10922ES20 |
| LDH Cytotoxicity Assay Kit | Yeasen Biotechnology, Shanghai, CN | Cat#40209ES76 |
| Hifair III 1st Strand cDNA Synthesis SuperMix | Yeasen Biotechnology, Shanghai, CN | Cat#11141ES10 |
| Hieff qPCR SYBR Green Master Mix | Yeasen Biotechnology, Shanghai, CN | Cat#11201ES03 |

## Plasmids and reagents

The pmC1-TDP-43-Cherry, VR1012-TDP-43-N-HA, and pshRNA-TDP-43 plasmids were purchased from Generay Biotech Co. Ltd. (Shanghai, CN). The HIV-1-ΔEnv-EGFP vector (pNL4-3-ΔEnv-EGFP, Cat#11100) provided by Dr. Robert Siliciano (Johns Hopkins University) was obtained through the NIH AIDS Research and Reference Reagent Program. The expression vectors HA-MVP5180/YBF30-Vpu and HA-SIVcpz_TAN1-Vpu were synthesized by Sangon Biotech (Shanghai, CN). Replication-defective pseudovirus DHIV plasmids were gifts from Vicente Planelles (Bonczkowski et al, 2014). The lentivirus packaging plasmids pCMV-VSV-G (8454), pRSVRev (12253), and pMDLg/pRRE (12251) were purchased from Addgene. VR1012-TDP-43-CTF-HA was constructed with a Hieff Clone Universal One Step Cloning Kit (10922ES20, Yeasen Biotechnology, Shanghai, CN). The pmC1-TDP-43-D89E/D219E plasmids were constructed from pmC1-TDP-43-Cherry by site-directed mutagenesis.

MG132 (HY-13259), chloroquine (CQ) (HY-17589A), Z-VAD-FMK (HY-16658B), and Z-DEVD-FMK (HY-12466) were purchased from MedChemExpress. Recombinant human TNFα protein (Abcam, ab9642) and cycloheximide (CHX) (SelleckChem, S7418) were used. The following antibodies were used: anti-TDP-43 (Abcam, 104223), anti-total TDP-43 (Proteintech, 10782-2-AP), anti-TDP-43 C-terminal (Proteintech, 12892-1-AP), anti-Vpu (Abcam, 81532), anti-Caspase 3 (CST, 9662), anti-HA (Covance, MMS-101R-1000), anti-Myc (Abcam, ab32), anti-alpha tubulin (Abcam, ab7291), anti-GGA3 (Abcam, ab180951) and anti-histone H3 (Abcam, ab176842). The anti-CA p24 antibody (Abcam, ab9701) was also used. The monoclonal anti-CA p24 antibody (Cat #1513) was obtained from the AIDS Research and Reference Reagents Program.

## Cells

HEK293T and T98G cells were maintained in Dulbecco's modified Eagle's medium supplemented with 10% fetal bovine serum and penicillin/streptomycin. THP-1 and Jurkat cells were maintained in Roswell Park Memorial Institute (RPMI) 1640 medium supplemented with 10% fetal bovine serum and penicillin/streptomycin. SH-SY5Y cells were maintained in Eagle's minimum essential medium/F12 supplemented with 10% fetal bovine serum, 1% nonessential amino acids (NEAAs), and penicillin/streptomycin. Primary human astrocytes (from a 50-year-old Chinese male donor) were maintained in complete culture medium (PriMed-iCell-007, iCell Bioscience Inc.). All cultured cells were maintained at 37 °C in a humidified atmosphere containing 5% $CO_2$. All cell lines have been STR verified and confirmed to be mycoplasma free at the time of use.

## Viruses and infection

For HIV-1-EGFP-VSV-G viral production, HEK293T cells in 100-mm dishes were transfected with 9 μg of pNL4-3-ΔEnv-EGFP and 1 μg of pCMV-VSV-G; for shRNA lentivirus production, HEK293T cells were transfected with 3 μg of the shRNA construct, 3 μg of pRSVRev, 3 μg of pMDLg/pRRE and 1 μg of pCMV-VSV-G. The supernatants were harvested 48 h later and then filtered at 0.45 μm. The lentivirus was purified and concentrated by ultracentrifugation at 28,000 rpm for 2 h with a 20% sucrose cushion. The virus pellet was then dissolved in PBS and stored at −80 °C. For single infection with the HIV-1-EGFP-VSV-G virus, the cells were infected with FBS-free medium, and replaced with DMEM containing 10% FBS 2 h later. The efficiency of productive infection was analyzed 48 h later by flow cytometry (FACSCalibur, BD Biosciences). All the results are representative of three biological replicates.

## Immunoblotting

Cell samples were harvested by scraping, washed with cold PBS, lysed in lysis buffer (150 mM Tris, pH 7.5, with 150 mM NaCl, 1% Triton X-100, and complete protease inhibitor cocktail tablets [Roche]) at 4 °C for 30 min, and centrifuged at 10,000 × g for 30 min. The supernatants were mixed with 1× loading buffer (0.08 M Tris, pH 6.8, with 2.0% SDS, 10% glycerol, 0.1 M DTT, and 0.2% bromophenol blue), boiled for 10 min and then centrifuged at 12,000 rpm for 10 min. The cell lysates were separated via SDS–PAGE and transferred to nitrocellulose membranes using a semidry apparatus (Bio-Rad). The membranes were probed with various primary antibodies against the proteins of interest as previously described.

## Immunofluorescence

The cells were washed twice with PBS at 48 h after transfection, fixed with 4% paraformaldehyde for 30 min at room temperature, permeabilized in 0.3% Triton X-100 for 10 min, and blocked in a 5% BSA solution for 1 h. The cells were subsequently incubated with an antibody overnight at 4 °C. The cells were incubated with goat anti-rabbit Alexa 488 (Life Technologies, A-32731) or goat anti-mouse Alexa 594 (Life Technologies A-11005) for 1 h at 4 °C. Nuclei were counterstained with 4′,6-diamidino-2-phenylindole (DAPI). Fluorescence imaging was performed using a confocal microscope (Nikon AX) with a ×100 oil objective and acquired using NIS-Elements AR 5.41.01.

## Cell fractionation

Proteins were sequentially extracted using RIPA and urea buffers to examine the solubility of TDP-43 (Winton et al, 2008). The cells were washed twice with PBS and lysed with cold RIPA buffer (50 mM Tris-HCl, 150 mM NaCl, 1% NP-40, and 2.5 mM EDTA, containing complete protease inhibitor cocktail tablets [Roche] and PMSF). For total protein from the cells, the cell lysates were sonicated (20 kHz, 500 W, 30% amplitude, on 3 s, off 3 s, 10 cycles) and centrifuged at 17,000 rpm for 30 min at 4 °C. The supernatants were RIPA-soluble fractions, while the pellets were RIPA-insoluble fractions. After washing by resonication and recentrifugation, the RIPA-insoluble pellets were extracted with 7 M urea buffer (7 M urea, 2 M thiourea, 4% CHAPS, and 30 mM Tris, pH 8.5). For the separation of nuclear and cytoplasmic proteins, the cells were lysed with RIPA buffer and incubated on ice for 5–10 min. When the suspension was clear, the samples were centrifuged at 500 × g for 10 min at 4 °C. The supernatants were carefully and gently aspirated to obtain the cytoplasmic fraction. The nuclear pellet was washed with ice-cold RIPA buffer and recentrifuged at 500 × g

for 10 min at 4 °C. The pellet was collected and resuspended in RIPA buffer as the nuclear fraction. The cytoplasmic fraction and nuclear fraction were sonicated and centrifuged as described above to separate the soluble and insoluble RIPA fractions.

## Toxicity assay

LDH levels were measured with an LDH Cytotoxicity Assay Kit (40209ES76, Yeasen Biotechnology, Shanghai, CN). Briefly, the supernatants were centrifuged at 1000 rpm for 10 min 72 h post transfection, diluted tenfold with PBS and incubated with a working LDH solution in the dark for 30 min at room temperature. The absorbance at 490/600 nm was measured with a microplate reader (Bio-Rad).

## Crystal violet staining

The cells in the 12-well plates were washed twice with PBS and then fixed with methyl alcohol for 30 min at room temperature. After being stained with a 0.5% crystal violet solution for 1 h, the cells were rinsed with pure water and left to dry for imaging.

## Quantitative real-time PCR

Total RNA from cells was isolated using TRIzol (Life Technologies) according to the manufacturer's instructions. The RNA was reverse transcribed using Hifair III 1st Strand cDNA Synthesis SuperMix for qPCR (YEASEN,11141ES10). Hieff qPCR SYBR Green Master Mix (YEASEN, 11201ES03) was used for qRT–PCR amplification, and the primers used are listed in Appendix Table S1.

## Statistical analyses

The investigators were not blinded to experimental groups while collecting data. All the statistical analyses were performed using GraphPad Prism software (version 5.0). Statistical analyses were performed using one-way ANOVA. A value of $P < 0.05$ was considered to indicate statistical significance ($*P < 0.05$; $**P < 0.01$; $***P < 0.001$; $****P < 0.0001$). All experiments were performed with at least three biological replicates.

## Biosafety

HIV-1-based pseudovirus experiments were performed in biosafety laboratory level 2 (BSL-2) and approved by the Insititute of Virology and AIDS Research, First Hospital of Jilin University and conducted in compliance with biosafety guidelines.

# Data availability

This study includes no data deposited in external repositories.

The source data of this paper are collected in the following database record: biostudies:S-SCDT-10_1038-S44319-024-00238-y.

# Peer review information

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

## Acknowledgements

The authors thank Dr. Tan Xu for the professional advice and Ling Xue, Yuanyuan Li for technical assistance. This work was supported by the NSFC Excellent Young Scientist Fund (32222005), the National Natural Science Foundation of China (82372226, 82172246, 92169203, 31970151), the National Major Project for Infectious Disease Control and Prevention (2018ZX10731-101-001-016), the Department of Science and Technology of Jilin Province (No. 20210101015JC), the Open Project of Key Laboratory of Organ Regeneration and Transplantation, Ministry of Education, the Program for JLU Science and Technology Innovative Research Team (2017TD-08), and Fundamental Research Funds for the Central Universities.

## Author contributions

**Jiaxin Yang**: Data curation; Formal analysis; Validation; Investigation; Methodology; Writing—original draft; Writing—review and editing. **Yan Li**: Formal analysis; Validation; Methodology. **Huili Li**: Formal analysis; Validation; Investigation. **Haichen Zhang**: Formal analysis; Validation. **Haoran Guo**: Formal analysis; Methodology. **Xiangyu Zheng**: Formal analysis. **Xiao-Fang Yu**: Formal analysis. **Wei Wei**: Conceptualization; Supervision; Investigation; Writing—original draft; Project administration; Writing—review and editing.

Source data underlying figure panels in this paper may have individual authorship assigned. Where available, figure panel/source data authorship is listed in the following database record: biostudies:S-SCDT-10_1038-S44319-024-00238-y.

## Disclosure and competing interests statement

The authors declare no competing interests.

# Expanded View Figures

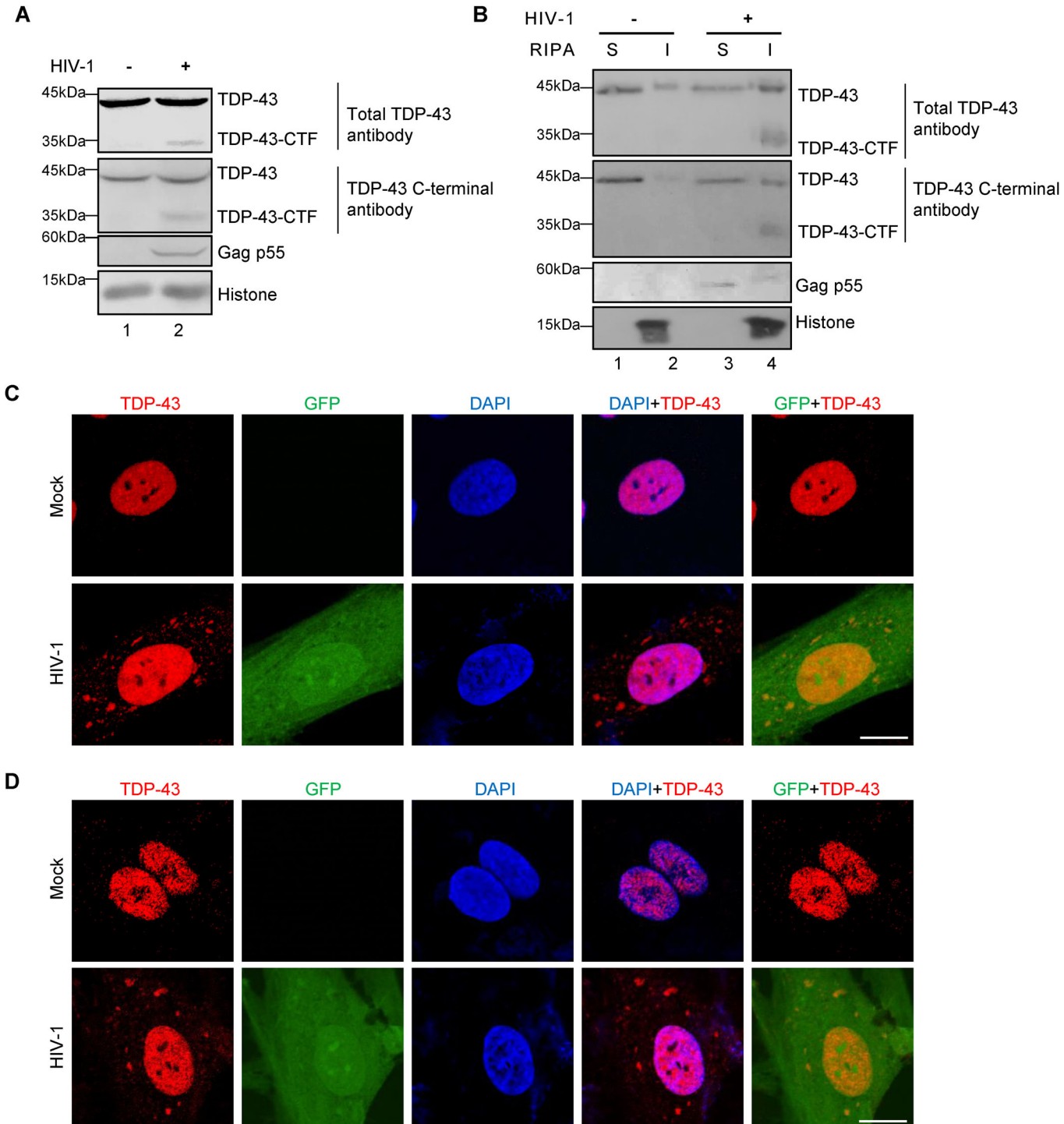

**Figure EV1.  HIV-1 induces the cleavage and cytoplasmic aggregation of TDP-43.**

(**A**) Western blotting of primary human astrocyte cells infected with the HIV-1-ΔEnv-EGFP-VSV-G virus. TDP-43 antibodies were used to detect endogenous TDP-43. CTF, C-terminal fragment. (**B**) Cell fractionation analysis of SH-SY5Y cells infected with the HIV-1-ΔEnv-EGFP-VSV-G virus. S soluble, I insoluble. (**C, D**) Immunofluorescence images of primary human astrocyte cells infected with the HIV-1-ΔEnv-EGFP-VSV-G virus. Total TDP-43 antibody (Proteintech, 10782-2-AP) (**C**) and TDP-43 (C-terminal) antibody (Proteintech, 12892-1-AP) (**D**) were used to detect TDP-43. Scale bar, 10 μm.

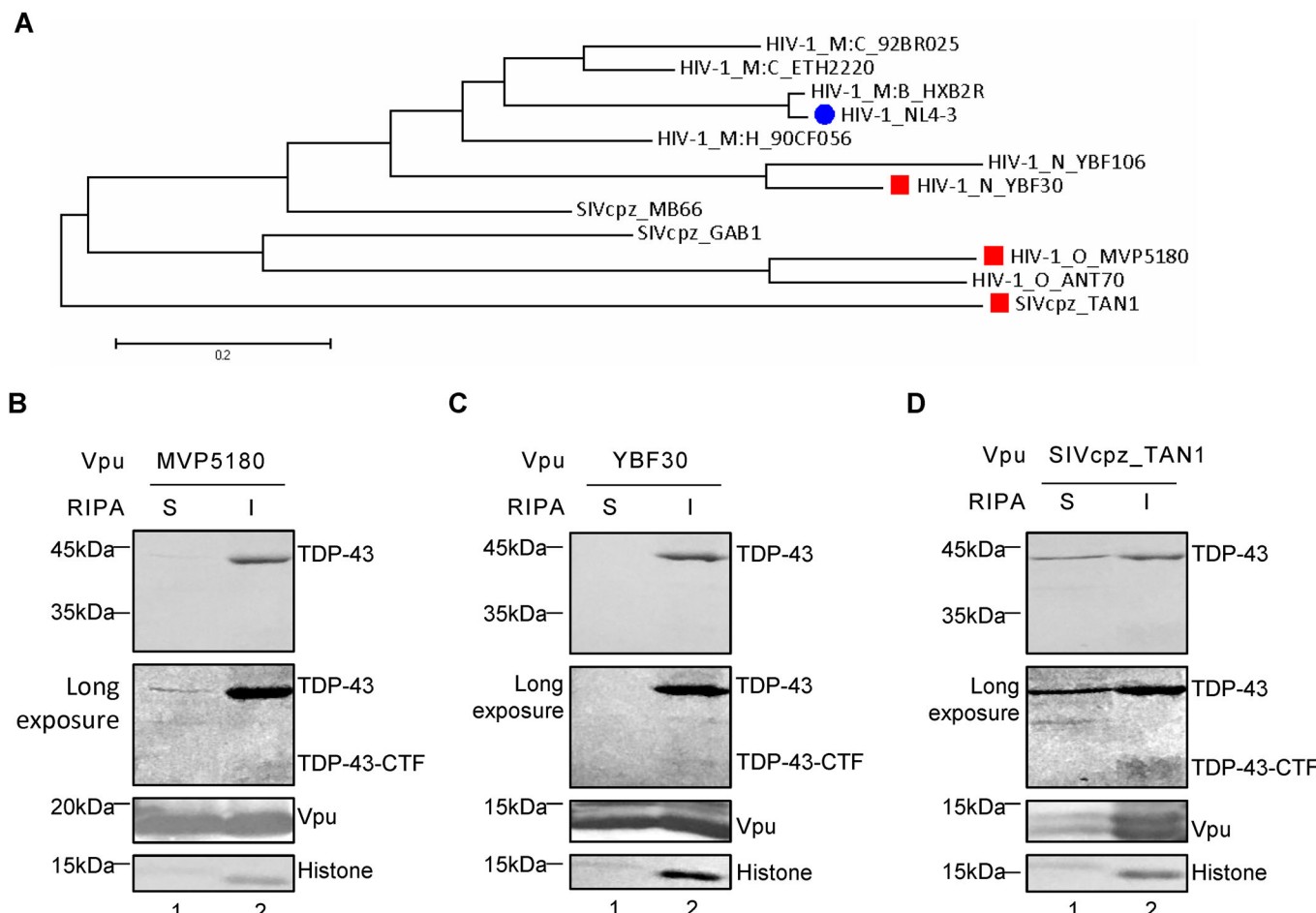

**Figure EV2. The aggregation of TDP-43 is induced by Vpu derived from different strains of HIV-1 and SIVcpz_TAN1.**

(A) The amino acid sequences of the indicated HIV-1/SIV Vpu proteins were retrieved from the HIV database (www.hiv.lanl.gov). Evolutionary history was inferred using the neighbor-joining method, and evolutionary analyses were conducted in MEGA7. The tree is drawn to scale, with branch lengths and evolutionary distances used to infer the phylogenetic tree shown in the same units. (B–D) Cell fractionation analysis of HEK293T cells transfected with expression plasmids as shown. S soluble, I insoluble. The asterisk indicates the cleavage product of TDP-43.

**A**

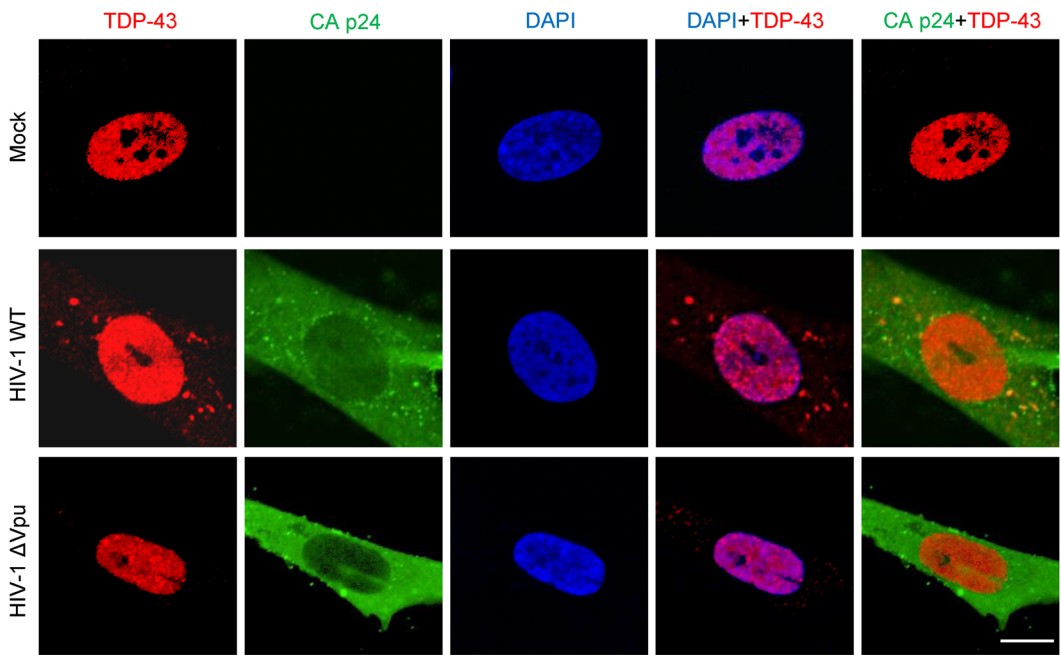

**B**

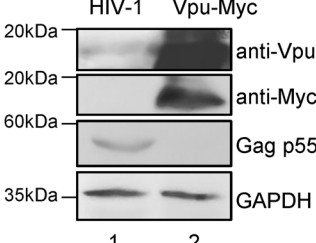

Figure EV3. **Vpu is essential for the cytoplasmic translocation of TDP-43 induced by HIV-1.**

(A) Immunofluorescence images of primary human astrocyte cells infected with the HIV-1-ΔEnv-VSV-G virus. CA p24 antibodies were used to detect HIV-1. Scale bar, 10 μm. (B) Western blotting of HEK293T cells infected with the HIV-1-ΔEnv-EGFP-VSV-G virus or transfected with VR1012-Vpu-Myc. CA p24 antibody was used to detect HIV-1 infection. A Myc-tag antibody was used to detect Vpu overexpression. A Vpu antibody was used to detect Vpu protein in all the samples.

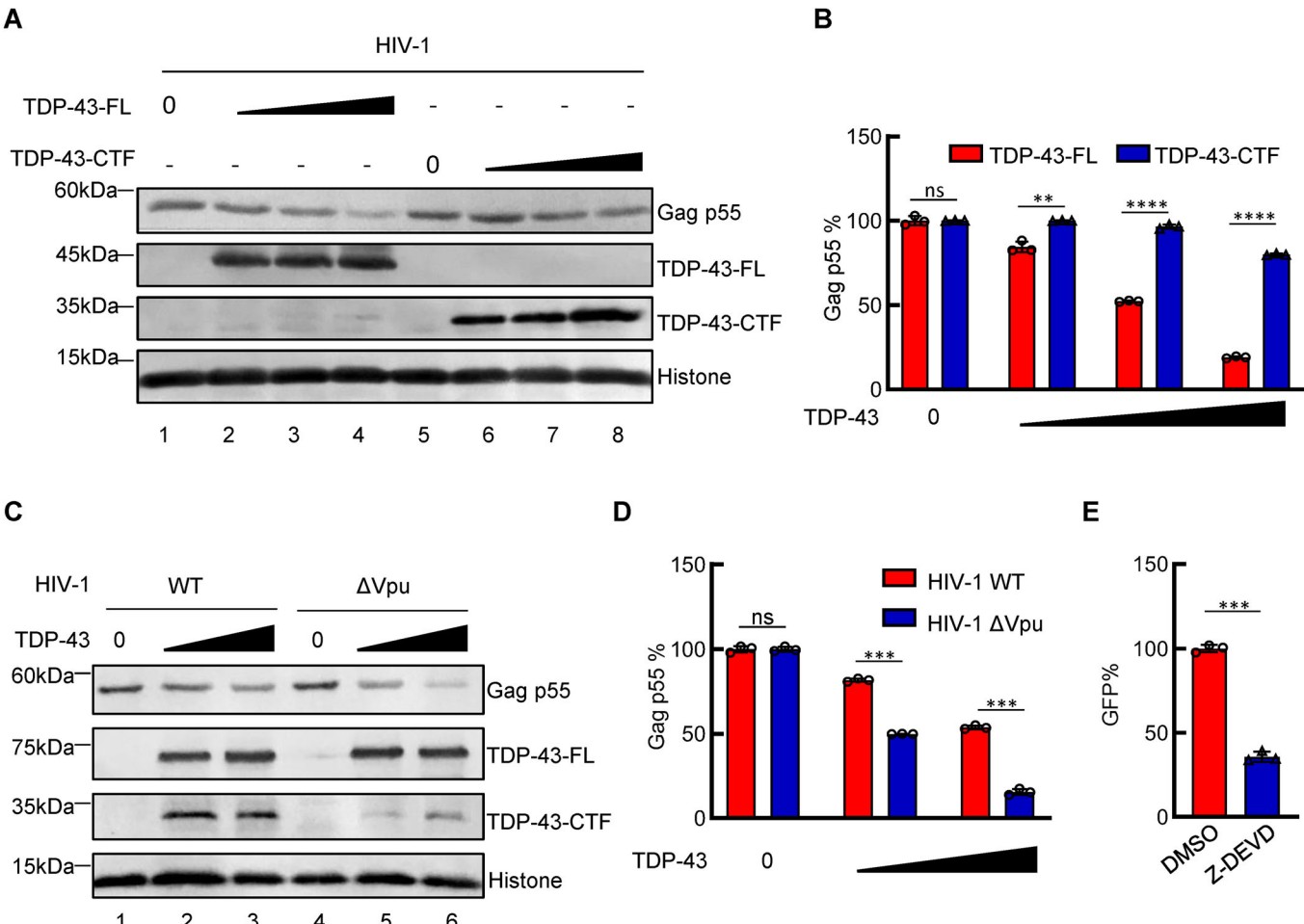

**Figure EV4.   The cleavage of TDP-43 is facilitated by Vpu to augment HIV-1 transcription.**

(A, B) Western blotting of HEK293T cells cotransfected with HIV-1-ΔEnv-EGFP and different doses of the VR1012-TDP-43-FL-HA- or VR1012-TDP-43-CTF-HA expression vectors (0, 0.25, 0.5, or 1 µg). A CA p24 antibody was used to detect Gag p55 in cell lysates. HA-tag antibodies were used to detect TDP-43-FL and TDP-43-CTF. The bar graph shows the relative percentage of Gag p55 according to grey analysis, ns not significant, **P = 0.0012, ****P < 0.0001. (C, D) HEK293T cells were cotransfected with wild-type or ΔVpu HIV-1-ΔEnv plasmids and a low level of pmC1-TDP-43 (0, 0.2 or 0.4 µg). The cells were collected for Western blotting with the indicated antibodies. The relative levels of Gag p55 according to grey analysis are shown in the bar graph. ImageJ was used for grey analysis, ns not significant, 0.2 µg group ***P = 0.0003, 0.4 µg group ***P = 0.0018. (E) The infection efficiency of the HIV-1-ΔEnv-EGFP-VSV-G virus was detected by flow cytometry. The HIV-1-ΔEnv-EGFP-VSV-G virus was packaged in HEK293T cells, and then the cells were treated with DMSO or Z-DEVD-FMK (20 µM) posttransfection. The supernatants were collected for infection, and infectivity was measured 48 h after viral infection using flow cytometry to detect GFP-positive cells, ***P = 0.0007. Data information: data are presented as mean ± SEM. ANOVA, n = 3 biological replicates.

   

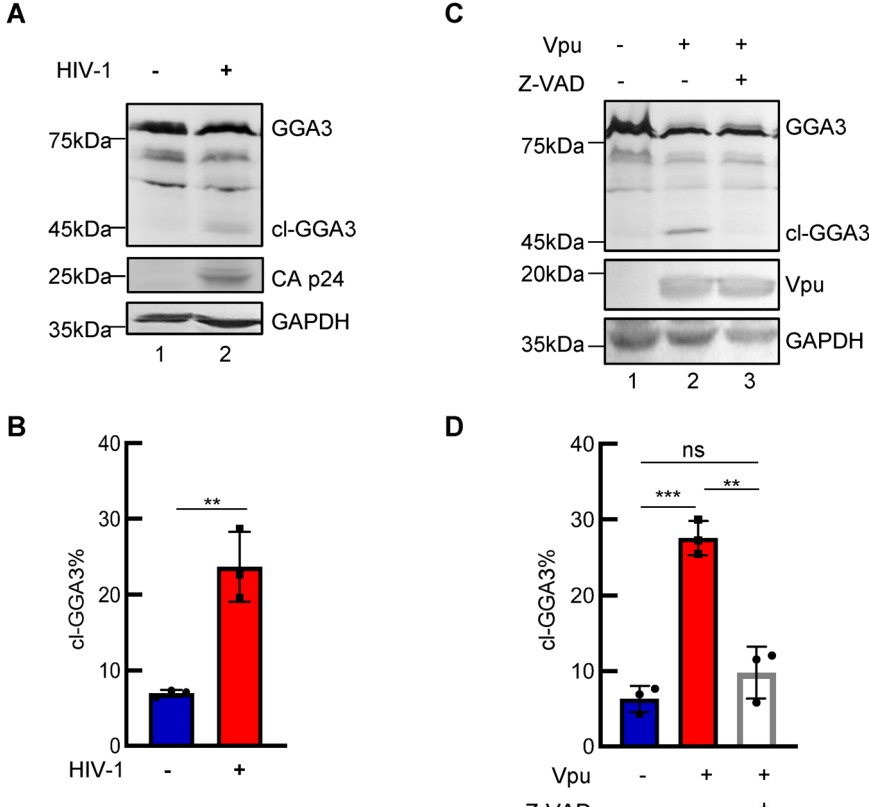

**Figure EV5. HIV-1 Vpu induces cleavage of GGA3 in a Caspase-dependent manner.**

(A) Western blotting of HEK293T cells transfected with the HIV-1-ΔEnv-EGFP plasmid to detect endogenous GGA3 using a GGA3 antibody. (B) The bar graph shows the percentages of the relative band intensities for cl-GGA3 relative to total GGA3 according to grey analysis. **$P = 0.0034$. (C) Western blotting of HEK293T cells transfected with VR1012-Vpu-Myc and treated with DMSO or 20 µM Z-VAD-FMK at 4 h posttransfection. cl, cleaved. (D) The bar graph shows the percentages of the relative band intensities for cl-GGA3 relative to total GGA3 according to grey analysis. ***$P = 0.0002$, **$P = 0.0017$, ns not significant. Data information: data are presented as mean ± SEM. ANOVA, $n = 3$ biological replicates.

