## [Peer Review File · EMBO Reports]

HIV-1 Vpu induces neurotoxicity by promoting Caspase 3-dependent cleavage of TDP-43

Jiaxin Yang, Yan Li, Huili Li, Haichen Zhang, Haoran Guo, Xiangyu Zheng, Xiaofang Yu, and Wei Wei

Corresponding author: Wei Wei (wwei6@jlu.edu.cn)

Review Timeline:

Submission Date:	5th Mar 24
Editorial Decision:	15th Apr 24
Revision Received:	17th Jul 24
Editorial Decision:	8th Aug 24
Revision Received:	9th Aug 24
Accepted:	13th Aug 24

Editor: Achim Breiling

Transaction Report:

Dear Prof. Wei,

Thank you for the submission of your manuscript to EMBO reports. I have now received the reports from the three referees that were asked to evaluate your study, which can be found at the end of this email.

As you will see, all referees have several comments, concerns, and suggestions, indicating that a major revision of the manuscript is necessary to allow publication of the study in EMBO reports. As the reports are below, and all the concerns need to be addressed, I will not detail them here.

Given the constructive referee comments, I would like to invite you to revise your manuscript with the understanding that the concerns of the referees must be addressed in the revised manuscript or in a detailed point-by-point response. Acceptance of your manuscript will depend on a positive outcome of a second round of review. It is EMBO reports policy to allow a single round of revision only and acceptance of the manuscript will therefore depend on the completeness of your responses included in the next, final version of the manuscript.

- 1) a .docx formatted version of the final manuscript text (including legends for main figures, EV figures and tables), but without the figures included. Figure legends should be compiled at the end of the manuscript text.
- 2) individual production quality figure files as .eps, .tif, .jpg (one file per figure), of main figures and EV figures. Please upload these as separate, individual files upon re-submission.

The Expanded View format, which will be displayed in the main HTML of the paper in a collapsible format, has replaced the Supplementary information. You can submit up to 5 images as Expanded View. Please follow the nomenclature Figure EV1, Figure EV2 etc. The figure legend for these should be included in the main manuscript document file in a section called Expanded View Figure Legends after the main Figure Legends section. Additional Supplementary material should be supplied as a single pdf file labelled Appendix. The Appendix should have page numbers and needs to include a table of content on the first page (with page numbers) and legends for all content. Please follow the nomenclature Appendix Figure Sx, Appendix Table Sx etc. throughout the text, and also label the figures and tables according to this nomenclature.

- 4) a complete author checklist, which you can download from our author guidelines (<https://www.embopress.org/page/journal/14693178/authorguide>). Please insert page numbers in the checklist to indicate where the requested information can be found in the manuscript. The completed author checklist will also be part of the RPF.

- 5) that primary datasets produced in this study (e.g. RNA-seq, ChIP-seq, structural and array data) are deposited in an

appropriate public database. If no primary datasets have been deposited, please also state this in a dedicated section (e.g. 'No primary datasets have been generated and deposited'), see below.

The accession numbers and database should be listed in a formal "Data Availability" section (placed after Materials & Methods) that follows the model below. This is now mandatory (like the COI statement). Please note that the Data Availability Section is restricted to new primary data that are part of this study. This section is mandatory. As indicated above, if no primary datasets have been deposited, please state this in this section

Data availability

7) Our journal encourages inclusion of *data citations in the reference list* to directly cite datasets that were re-used and obtained from public databases. Data citations in the article text are distinct from normal bibliographical citations and should directly link to the database records from which the data can be accessed. In the main text, data citations are formatted as follows: "Data ref: Smith et al, 2001" or "Data ref: NCBI Sequence Read Archive PRJNA342805, 2017". In the Reference list, data citations must be labelled with "[DATASET]". A data reference must provide the database name, accession number/identifiers and a resolvable link to the landing page from which the data can be accessed at the end of the reference. Further instructions are available at: <http://www.embopress.org/page/journal/14693178/authorguide#referencesformat>

8) Regarding data quantification and statistics, please make sure that the number "n" for how many independent experiments were performed, their nature (biological versus technical replicates), the bars and error bars (e.g. SEM, SD) and the test used to calculate p-values is indicated in the respective figure legends (also for EV figures and all those in an Appendix). Please also check that all the p-values are explained in the legend, and that these fit to those shown in the figure. Please provide statistical testing where applicable. Please avoid the phrase 'independent experiment', but clearly state if these were biological or technical replicates. Please also indicate (e.g. with n.s.) if testing was performed, but the differences are not significant. In case n=2, please show the data as separate datapoints without error bars and statistics. See also: <http://www.embopress.org/page/journal/14693178/authorguide#statisticalanalysis>

9) Please add scale bars of similar style and thickness to microscopic images, using clearly visible black or white bars (depending on the background). Please place these in the lower right corner of the images themselves. Please do not write on or near the bars in the image but define the size in the respective figure legend.

10) Please also note our reference format:

12) We now use CRedit to specify the contributions of each author in the journal submission system. CRedit replaces the author contribution section. Please use the free text box to provide more detailed descriptions and do not provide your final manuscript text file with an author contributions section. See also our guide to authors: <https://www.embopress.org/page/journal/14693178/authorguide#authorshippinguidelines>

13) We would encourage you to use 'Structured Methods', our new Materials and Methods format. According to this format, the

Materials and Methods section should include a Reagents and Tools Table (listing key reagents, experimental models, software, and relevant equipment and including their sources and relevant identifiers), uploaded as separate file, followed by a Methods and Protocols section in which we encourage the authors to describe their methods using a step-by-step protocol format with bullet points, to facilitate the adoption of the methodologies across labs. More information on how to adhere to this format as well as downloadable templates (.doc or .xls) for the Reagents and Tools Table can be found in our author guidelines (section 'Structured Methods'):

14) Please add up to five keywords to the manuscript and provide the abstract written in present tense. Please also order the manuscript sections like this, using these names:

Title page - Abstract - Keywords - Introduction - Results - Discussion - Methods - Data availability section - Acknowledgements - Disclosure and Competing Interests Statement - References - Figure legends - Expanded View Figure legends

I look forward to seeing a revised version of your manuscript when it is ready. Please let me know if you have questions or comments regarding the revision.

Yours sincerely,

Referee #1:

In this article, Yang and colleagues demonstrated that HIV-1 infection induces neurotoxicity by the action of the HIV-1 accessory protein Vpu, which activated Caspase-3. They further showed that Vpu ectopic expression resulted in TDP43 cleavage, and inhibition of Caspase-3 attenuated HIV-1 mediated neurotoxicity.

Overall, the novelty of this work is limited. TDP43 is known to be cleaved by caspases and this work marginally increases our knowledge of viral-induced neurotoxicity or TDP-43 mediated neurotoxicity.

Major comments:

Figure 1. The protein ladder should be included in the western blots. what is the size of the TDP43 CTFs?

Figure 2I. Many times in dividing cells, TDP43 is transiently localized to the cytoplasm. This is the case for the cells shown here. This reviewer is not convinced that the Vpu+ panels where all TDP43 is mislocalized shows a correct representation of the entire well. See also Figure 2J where only 50% of cells has cytoplasmic TDP43. Please replace with a more representative set of images.

3. Are there accumulation of cryptic exons with the loss of nuclear TDP43?

4. The TDP43 antibody used in this study is not ideal because the immunogen is unknown. It is recommended that the authors use a number of different TDP43 antibodies widely used in the field:

- TDP43 C-terminal antibody (Proteintech 12892-1-AP)
- Total TDP43 antibody (Proteintech 10782-2-AP)
- Phospho-TDP43 (22309-1-AP)

Referee #2:

The manuscript shows that the HIV-1 Vpu protein results in cleavage of TDP-43, translocation into the nucleus and aggregation, providing a further explanation for HIV-1-induced neurological damage.

While several published works, including those by this group, have shown viral proteases to cause the formation of TDP-43 aggregation, this has not been investigated for HIV. Furthermore, in this case, cleavage of TDP-43/ mis-localization and aggregation would seem to be due to Vpu activation of Caspase 3.

The conclusions drawn from the experimental data are reasonable.

Two points merit further investigation.

1. An aspect of TDP-43 proteinopathy not fully investigated is its loss of function. Is the amount of soluble TDP-43 enough to fulfil cellular functions?

For example; the mislocalization of TDP-43 in cells overexpressing Vpu is extensive, while for example in Figure 1 in the HIV-infected astrocyte cells that would express the Vpu protein at a more physiological level, this is less dramatic. Notwithstanding this, the change in the ratio of soluble to insoluble TDP-43 is substantial and more than would have been expected. It may be the case that nuclear aggregation of TDP-43 is also occurring. To test this the authors could also perform the solubility experiments on nuclear and cytoplasmic fractions. More importantly, however, the authors should investigate if there is a loss of TDP-43 function in cell lines infected with HIV, for example through analysis of known splicing targets of TDP-43. This would add a further layer of information regarding the effect of Vpu caspase activation and would significantly reinforce the manuscript's conclusions. In other words, the authors should investigate if, in the presence of the toxicity associated with the cleaved fragments and the mis-localization, there is also a loss of TDP-43 function in an HIV infection.

2. Can any evidence that cytoplasmic accumulation follows cleavage be provided? For example: HIV-1ΔVpu infection, does this result in a difference in cytoplasmic/nuclear TDP-43? Can the caspase inhibitors used in the HIV-infected cells and microscopy be performed or fractionation of the nuclear and cytoplasmic fractions be performed to evaluate the state of TDP-43 localization aggregation etc in the absence of the cleaved fragments but in an HIV infection background?

Minor corrections:

1. line 56-58- the authors should clarify that TDP-43 is not a pathological protein per se but its aggregation, reduced functionality, aberrant expression, mis-localization etc are associated with various neurological disorders. As the sentences stands this is not clear.

2. Caspase 3 activation has been seen to occur with several other components of HIV (i.e envelope proteins). In fact, Vpr whose overexpression did not result in cleaved TDP-43 fragments, has also been shown to activate Caspase 3. These aspects should be discussed.

3. No statistical significance is shown in the bar charts of 3F.

4. Was a comparison between the Vpu levels from the expression plasmid and from HIV infections made?

Referee #3:

TAR DNA-binding protein (TDP-43). is a cellular protein with crucial roles in RNA splicing, transcription, mRNA transport, and mRNA stabilization. Here, the authors find that HIV-1 infection can induce cleavage and aberrant aggregation of TDP-43. The authors find that the aggregation and cleavage is caused by caspase 3, which is activated by the Vpu protein. The authors identify residue D89 in the 414 amino acid TDP-43 protein as the caspase 3 cleavage site. A second potential caspase 3 motif is apparently not used. Cleaved TDP-43 is cytotoxic in neural cells and expression of the cleaved TDP-43 CTF has dominant negative properties. Altogether, the study identifies TDP-43 as a novel target of Vpu and expression of Vpu results in the caspase-3 mediated cleavage of TDP-43.

The data presented here are interesting and make a reasonable case that Vpu is involved in the caspase-3-mediated cleavage of TDP-43. However, caspase-3 does not selectively target TDP-43 but cleaves a number of other cellular proteins as part of its involvement in host cell apoptosis. Caspase-3 is an effector caspase whose activation requires cleavage by an initiator caspase (e.g. caspase-9), which itself is activated through a pro-apoptotic signaling cascade. The authors are obviously not aware of a 2001 study (Akari et al; PMID: 11696595) which reports that Vpu induces apoptosis by suppressing the nuclear factor kappa B-dependent expression of antiapoptotic factors. Akari et al demonstrate that Vpu does not directly activate caspase-3. Instead, Vpu inhibits the expression of anti-apoptotic genes which normally prevent activation of caspase-3. Vpu does this by interfering with the activation of NFkB. Akari et al did not identify or characterize specific targets of caspase-3. Thus, the current study is a logical extension of the prior work and should be discussed in the context of the 2021 study.

If cleavage of TDP-43 by caspase-3 is unrelated to the general induction of apoptosis by Vpu, then treatment of cells with TNFalpha should result in Vpu independent cleavage of TDP-43. This should be tested.

What are the cytoplasmic aggregates shown in HIV-1 infected astrocytes in Fig. 1. The authors do not reveal what proteins are recognized by their HIV-1 specific antibody. However, there does not appear to be any colocalization. Research done with Vif and APOBEC3G previously revealed the presence of high molecular weight RNA protein complexes that include Vif and A3G. The authors should try to stain with specific antibodies to see if these aggregates include Vpu.

Fig. 2I suggests that Vpu expression induced TDP-43 translocation into the cytoplasm. How come, we do not see cytoplasmic aggregates as in Fig. 1A?

The authors state in their introduction that TDP-43 under normal conditions is predominantly localized to the nucleus. I therefore

assume that in the Ctrl samples in Fig 1C, TDP-43 is largely nuclear; yet, the fractionation into RIPA soluble and insoluble fraction shows TDP-43 in both fractions. Furthermore, histone is a nuclear marker and accumulates in the insoluble fraction. Based on that, TDP-43 would be read as mostly cytoplasmic in the ctrl sample since it is more prominent in the soluble fraction. This is highly confusing. I think the authors need to provide a detailed description of their fractionation analysis. In particular, they need to explain the difference between their RIPA/Urea fractionation and commercial fractionation kits that differentiate cytoplasmic, membrane, nuclear, and post-nuclear fractions.

A major technical weakness of the study is that methodologies are described in a very superficial manner. For instance, a key assay of this study is the "widely used" (line 88) cell fractionation assay that employs a combination of RIPA buffer and 7M urea extraction. Yet, there is no information on the buffer composition or the details of the sonication procedure.

Protein gels are sliced ultrathin and in most instances bands for full length TDP-43 and the CTF fragment are cut apart even though they can and should be shown on the same gel.

Molecular weight standards are missing throughout.

Figure 1: p55gag is RIPA buffer insoluble. Is that really true? What about p24 capsid?

Fig 3C: Vpu reduces levels of TDP-43-HA as expected but Z-DEVD-FMK treatment does not restore levels. Why not?

Fig 4B: labels in the cartoon are difficult to read. Also, the only obvious common feature in the two caspase-3 recognition motifs is the aspartic acid residue that appears to be the cleavage site. How were these motifs defined (provide a reference)? Panel A suggests the presence of a 2nd aspartic acid residue upstream of the proposed cleavage site; however, the cartoon stops short of that (only shows VMDVFI).

Achim Breiling
Senior Editor
*EMBO Reports*

Dear Dr. Breiling,

Thank you very much for your letter of April 15 concerning our manuscript entitled "**HIV-1**
**Initiates Neurotoxicity Via the Vpu/Caspase 3/TDP-43 Axis**"(EMBOR-2024-59135V1).
We addressed each of the reviewers' points in the letter below and made relevant
changes in the manuscript.

We are very grateful for the high quality reviews. It is obvious that each reviewer read the
manuscript very carefully. We truly appreciate the feedback, which will substantially
improve the reproducibility, quality and clarity of our paper.

Reviewers' comments:

Referee #1:

In this article, Yang and colleagues demonstrated that HIV-1 infection induces
neurotoxicity by the action of the HIV-1 accessory protein Vpu, which activated Caspase-
3. They further showed that Vpu ectopic expression resulted in TDP43 cleavage, and
inhibition of Caspase-3 attenuated HIV-1 mediated neurotoxicity.

Overall, the novelty of this work is limited. TDP43 is known to be cleaved by caspases
and this work marginally increases our knowledge of viral-induced neurotoxicity or TDP-
43 mediated neurotoxicity.

**Response:** We thank the reviewer for the careful reading and constructive criticism. We
have taken these points into serious consideration and revised the manuscript
accordingly. Specific comments are also addressed below.

Major comments:

Figure 1. The protein ladder should be included in the western blots. what is the size of
the TDP43 CTFs?

**Response:** As suggested by the reviewer, we have incorporated the protein ladder
information into the revised figures (new Figures 1C, 1E, 1G, 1F-I, 2A-2H, 3A-3E, 4C-D,
5F, 5H). The TDP-43 C-terminal fragment (CTF) generated by HIV-1 Vpu has an
approximate molecular weight of 35 kDa (page 5, lines 98-99).

Figure 2I. Many times in dividing cells, TDP43 is transiently localized to the cytoplasm.
This is the case for the cells shown here. This reviewer is not convinced that the Vpu+
panels where all TDP43 is mislocalized shows a correct representation of the entire well.
See also Figure 2J where only 50% of cells has cytoplasmic TDP43. Please replace with
a more representative set of images.

**Response:** Thank you very much for this valuable suggestion. We have replaced the
figures with images at a lower magnification as recommended (new Figure 2I).

3. Are there accumulation of cryptic exons with the loss of nuclear TDP43?

**Response:** We sincerely appreciate the reviewer for pointing this out. We have
conducted the recommended experiments and demonstrated that HIV-1-induced
cytoplasmic aggregation of TDP-43 results in the accumulation of cryptic exons of
UNC13A, ATG4B and GSPM2. The data have been incorporated as new Appendix
Figure S6 in the revised manuscript (page 11, lines 300-308).

4. The TDP43 antibody used in this study is not ideal because the immunogen is
unknown. It is recommended that the authors use a number of different TDP43
antibodies widely used in the field:

- TDP43 C-terminal antibody (Proteintech 12892-1-AP)

- Total TDP43 antibody (Proteintech 10782-2-AP)

- Phospho-TDP43 (22309-1-AP)

**Response:** The provided suggestion is very helpful. We have successfully validated the
HIV-1-induced cleavage and cytoplasmic aggregation of TDP-43 using the
recommended antibodies (Total TDP43 antibody, Proteintech, 10782-2-AP and TDP43
C-terminal antibody, Proteintech 12892-1-AP), and we found that these antibodies are
indeed more sensitive than those we used previously. The new data have been
incorporated into the new Figure EV1 (page 5, lines 105-107).

-----
**Referee #2:**

The manuscript shows that the HIV-1 Vpu protein results in cleavage of TDP-43,
translocation into the nucleus and aggregation, providing a further explanation for HIV-1-
induced neurological damage.

While several published works, including those by this group, have shown viral proteases
to cause the formation of TDP-43 aggregation, this has not been investigated for HIV.
Furthermore, in this case, cleavage of TDP-43/ mis-localization and aggregation would
seem to be due to Vpu activation of Caspase 3.

The conclusions drawn from the experimental data are reasonable.

**Response:** We greatly appreciate your constructive suggestions and the recognition of
the potential importance of our work. We have taken these points into consideration and
revised the manuscript accordingly. Specific comments are addressed below.

-----
Two points merit further investigation.

1. An aspect of TDP-43 proteinopathy not fully investigated is its loss of function. Is the
amount of soluble TDP-43 enough to fulfil cellular functions?

For example; the mislocalization of TDP-43 in cells overexpressing Vpu is extensive,
while for example in Figure 1 in the HIV-infected astrocyte cells that would express the
Vpu protein at a more physiological level, this is less dramatic.

**Response:** Thank you very much for pointing this out. We have measured the
expression levels of Vpu in both Vpu overexpressed cells and HIV-1-infected cells. The
immunoblotting data indicated that the levels of Vpu proteins were much lower in the
latter group (refer to new Figure EV3B, page 6, lines 137-138).

-----
Notwithstanding this, the change in the ratio of soluble to insoluble TDP-43 is substantial
and more than would have been expected. It may be the case that nuclear aggregation
of TDP-43 is also occurring. To test this the authors could also perform the solubility
experiments on nuclear and cytoplasmic fractions.

**Response:** The reviewer is correct. As suggested by the reviewer, we have conducted
the recommended experiments and demonstrated the aggregation of both cytoplasmic
and nuclear TDP-43 proteins in the presence of HIV-1 infection. These data have been
incorporated into our revised manuscript as Appendix Figure S2 (page 5, lines 114-119).

-----
More importantly, however, the authors should investigate if there is a loss of TDP 43
function in cell lines infected with HIV, for example through analysis of known splicing
targets of TDP-43. This would add a further layer of information regarding the effect of

Vpu caspase activation and would significantly reinforce the manuscript's conclusions. In
other words, the authors should investigate if, in the presence of the toxicity associated
with the cleaved fragments and the mis-localization, there is also a loss of TDP-43
function in an HIV infection.

**Response:** We sincerely appreciate this valuable suggestion. We have conducted the
recommended experiments and demonstrated that HIV-1-induced TDP-43 dysfunction
leads to an increasing accumulation of cryptic exons within UNC13A, ATG4B and
GSPM2 (new Appendix Figure S6, page 11, lines 300-308).

2. Can any evidence that cytoplasmic accumulation follows cleavage be provided? For
example: HIV-1ΔVpu infection, does this result in a difference in cytoplasmic/nuclear
TDP-43? Can the caspase inhibitors used in the HIV-infected cells and microscopy be
performed or fractionation of the nuclear and cytoplasmic fractions be performed to
evaluate the state of TDP-43 localization aggregation etc in the absence of the cleaved
fragments but in an HIV infection background?

**Response:** We thank the reviewer for this suggestion. We have conducted the
recommended experiments and demonstrated that HIV-1ΔVpu infection or treatment
with the caspase inhibitor z-VAD impairs the cytoplasmic accumulation of TDP-43
proteins triggered by HIV-1 (new Figures 3G and EV3A, page 7, lines 158-160 and page
6, lines 135-137).

Minor corrections:

1. line 56-58- the authors should clarify that TDP-43 is not a pathological protein per se
but its aggregation, reduced functionality, aberrant expression, mis-localization etc are
associated with various neurological
disorders. As the sentences stands this is not clear.

**Response:** We have corrected the description in the revised manuscript (page 3, lines
58-60).

2. Caspase 3 activation has been seen to occur with several other components of HIV
(i.e envelope proteins). In fact, Vpr whose overexpression did not result in cleaved TDP-
43 fragments, has also been shown to activate Caspase 3. These aspects should be
discussed.

**Response:** As suggested by the reviewer, we have incorporated a relevant discussion in
the revised manuscript (page 10, lines 273-278).

3. No statistical significance is shown in the bar charts of 3F.

**Response:** We have incorporated indicators of statistical significance in the revised
Figure 3F.

4. Was a comparison between the Vpu levels from the expression plasmid and from HIV
infections made?

**Response:** As suggested by the reviewer, we have conducted the recommended
experiments and incorporated the immunoblotting data of Vpu expression as a new
Figure EV3B (page 6, lines 137-138).

-----

Referee #3:

TAR DNA-binding protein (TDP-43). is a cellular protein with crucial roles in RNA
splicing, transcription, mRNA transport, and mRNA stabilization. Here, the authors find

that HIV-1 infection can induce cleavage and aberrant aggregation of TDP-43. The
authors find that the aggregation and cleavage is caused by caspase 3, which is
activated by the Vpu protein. The authors identify residue D89 in the 414 amino acid
TDP-43 protein as the caspase 3 cleavage site. A second potential caspase 3 motif is
apparently not used. Cleaved TDP-43 is cytotoxic in neural cells and expression of the
cleaved TDP-43 CTF has dominant negative properties. Altogether, the study identifies
TDP-43 as a novel target of Vpu and expression of Vpu results in the caspase-3
mediated cleavage of TDP-43.

**Response:** We greatly appreciate the positive support from the reviewer and find the
comments both insightful and helpful. Below are our point-by-point responses to the
reviewer's critiques.

The data presented here are interesting and make a reasonable case that Vpu is
involved in the caspase-3-mediated cleavage of TDP-43. However, caspase-3 does not
selectively target TDP-43 but cleaves a number of other cellular proteins as part of its
involvement in host cell apoptosis. Caspase-3 is an effector caspase whose activation
requires cleavage by an initiator caspase (e.g. caspase-9), which itself is activated
through a pro-apoptotic signaling cascade. The authors are obviously not aware of a
2001 study (Akari et al; PMID: 11696595) which reports that Vpu induces apoptosis by
suppressing the nuclear factor kappa B-dependent expression of antiapoptotic factors.
Akari et al demonstrate that Vpu does not directly activate caspase-3. Instead, Vpu
inhibits the expression of anti-apoptotic genes which normally prevent activation of
caspase-3. Vpu does this by interfering with the activation of NFkB. Akari et al did not
identify or characterize specific targets of caspase-3. Thus, the current study is a logical
extension of the prior work and should be discussed in the context of the 2021 study.

**Response:** We thank the reviewer for this suggestion. We have incorporated the
relevant discussions into the revised manuscript (page 10, lines 270-273).

If cleavage of TDP-43 by caspase-3 is unrelated to the general induction of apoptosis by
Vpu, then treatment of cells with TNFalpha should result in Vpu independent cleavage of
TDP-43. This should be tested.

**Response:** The reviewer is correct. We have performed the suggested experiments and
demonstrated that TNFalpha/cycloheximide (CHX) treatment can induce Vpu-
independent TDP-43 cleavage and cytoplasmic aggregation (new Appendix Figure S4,
page 7, lines 161-164).

What are the cytoplasmic aggregates shown in HIV-1 infected astrocytes in Fig. 1. The
authors do not reveal what proteins are recognized by their HIV-1 specific
antibody. However, there does not appear to be any colocalization. Research done with
Vif and APOBEC3G previously revealed the presence of high molecular weight RNA
protein complexes that include Vif and A3G. The authors should try to stain with specific
antibodies to see if these aggregates include Vpu.

**Response:** --In Figure 1A, the green fluorescence represents the spatial distribution of
GFP proteins expressed by the HIV-1 reporter viruses. We have improved the labeling in
the revised Figure 1A to enhance clarity and precision.

--We thank the reviewer very much for this suggestion. We have conducted the
recommended experiments and determined that the Vpu protein did not colocalize with
TDP-43 aggregates, as observed by an immunofluorescence assay (new Appendix
Figure S3, page 6, lines 141-143).

Fig. 2I suggests that Vpu expression induced TDP-43 translocation into the cytoplasm.
How come, we do not see cytoplasmic aggregates as in Fig. 1A?

**Response:** To address this issue, we have improved the quality of the
immunofluorescent pictures in the revised Figure 2I.

The authors state in their introduction that TDP-43 under normal conditions is
predominantly localized to the nucleus. I therefore assume that in the Ctrl samples in Fig
1C, TDP-43 is largely nuclear; yet , the fractionation into RIPA soluble and insoluble
fraction shows TDP-43 in both fractions. Furthermore, histone is a nuclear marker and
accumulates in the insoluble fraction. Based on that, TDP-43 would be read as mostly
cytoplasmic in the ctrl sample since it is more prominent in the soluble fraction. This is
highly confusing. I think the authors need to provide a detailed description of their
fractionation analysis. In particular, they need to explain the difference between their
RIPA/Urea fractionation and commercial fractionation kits that differentiate cytoplasmic,
membrane, nuclear, and post-nuclear fractions.

**Response:** We thank the reviewer for pointing this out. In this study, we used the
RIPA/Urea fractionation assay, a commonly used method in the TDP-43 research field,
to distinguish between RIPA-soluble and RIPA-insoluble proteins within various cellular
contexts (Lg, Da Silva, et al. PMID:35112738, Winton, Matthew J. et al. PMID:18305110,
Neumann, Manuela, et al. PMID:17023659). Histone H3 was employed as a marker to
delineate the insoluble fraction (Audas,et al. PMID: 27720612). In contrast to
conventional subcellular fractionation methods that use relatively gentle lysis approaches
that preserve nuclear membrane integrity while disrupting cell membrane integrity, the
RIPA/Urea fractionation assay combines RIPA lysis and sonication for complete
disruption of both cell and nuclear membranes during lysis, followed by centrifugation to
isolate insoluble protein components as precipitates (pages 4-5, lines 92-94).

A major technical weakness of the study is that methodologies are described in a very
superficial manner. For instance, a key assay of this study is the "widely used" (line 88)
cell fractionation assay that employs a combination of RIPA buffer and 7M urea
extraction. Yet, there is no information on the buffer composition or the details of the
sonication procedure.

**Response:** As suggested by the reviewer, we have incorporated these specific details
into the revised manuscript (pages 14-15, lines 399-400,401,404-405).

Protein gels are sliced ultrathin and in most instances bands for full length TDP-43 and
the CTF fragment are cut apart even though they can and should be shown on the same
gel.

**Response:** As suggested by the reviewer, we have replaced these figures with unedited
pictures (new Figures 1E, 2A-2B, 3A-3C, and 4C-4D).

Molecular weight standards are missing throughout.

**Response:** We have incorporated the molecular weight standards into the revised
figures.

Figure 1: p55gag is RIPA buffer insoluble. Is that really true? What about p24 capsid?

**Response:** We thank the reviewer very much for this comment. We conducted solubility
analyses of p55gag and p24CA in different human cells, and observed that p24CA in all
tested cells was enriched mainly in the soluble fractions. However, in some cells, such

as T98G and HEK293T cells, p55gag proteins were mostly insoluble, whereas in SH-
SY5Y and primary astrocyte cells, p55gag proteins were much more soluble (new
Appendix Figure S1D-F). We noted that the differences in the solubility of p55gag in
different cells did not influence the ability of HIV-1 infection to reduce TDP-43 solubility
(page 5, lines 111-113).

Fig 3C: Vpu reduces levels of TDP-43-HA as expected but Z-DEVD-FMK treatment does
not restore levels. Why not?

**Response:** We have consistently observed that the expression of Vpu can
indiscriminately hinder ectopic protein expression but has no influence on endogenous
protein expression, and this effect is not dependent on the activation of Caspase 3 by
Vpu. We hypothesize that Vpu may exert an inhibitory effect on the functionality of
elements within the expression vector, such as the promoter. To address this issue, we
have included a relevant discussion in the revised manuscript (page 6, lines 153-156).

Fig 4B: labels in the cartoon are difficult to read.

**Response:** We have corrected this in the revised Figure 4B.

Also, the only obvious common feature in the two caspase-3 recognition motifs is the
aspartic acid residue that appears to be the cleavage site. How were these motifs
defined (provide a reference)?

**Response:** We have added the related citations to the revised manuscript (page 7, lines
166-167).

Panel A suggests the presence of a 2nd aspartic acid residue upstream of the proposed
cleavage site; however, the cartoon stops short of that (only shows VMDVFI).

**Response:** As suggested by the reviewer, we have corrected this issue in the revised
Figure 4B.

Dear Prof. Wei

Thank you for the submission of your revised manuscript to our editorial offices. I have now received the reports from two of the three referees that I asked to re-evaluate the study, you will find below. Referee #1 declined to look into the revised manuscript but going through your p-b-p-response and the revised manuscript, I consider her/his points as adequately addressed. As you will see, the other two referees now fully support the publication of the study in EMBO reports.

Before we can proceed with formal acceptance, I have these editorial requests I ask you to address in a final revised manuscript:

- I would suggest this more comprehensive title:

HIV-1 Vpu induces neurotoxicity by promoting Caspase 3-dependent cleavage of TDP-43

- Please order the manuscript sections like this, using these names:

Title page - Abstract - Keywords - Introduction - Results - Discussion - Methods - Data availability section - Acknowledgements - Disclosure and Competing Interests Statement - References - Figure legends - Expanded View Figure legends

- Please make sure that the number "n" for how many independent experiments were performed, their nature (biological versus technical replicates), the bars and error bars (e.g. SEM, SD) and the test used to calculate p-values is indicated in the respective figure legends (main, EV and Appendix figures). Please also check that all the p-values are explained in the legend, and that these fit to those shown in the figure. Please provide statistical testing where applicable. Please avoid the phrase 'independent experiment', but clearly state if these were biological or technical replicates. Please also indicate (e.g. with n.s.) if testing was performed, but the differences are not significant. In case n=2, please show the data as separate datapoints without error bars and statistics. See also:

<http://www.embopress.org/page/journal/14693178/authorguide#statisticalanalysis>

If n<5, please show single datapoints for diagrams. It seems that presently some diagrams have no or only partial stats or the 'n.s.' is missing. Moreover:

- Please note that the legends for figures EV 5b-c is not provided in the sequential manner (legend for figure 5c is provided before legend of figure 5b). This needs to be rectified.

- Please note that the exact p values are not provided in the legends of figures 1d; 2h; 3f; 4e; 5e, g, i; 6a, e.

- Please note that in figure EV 4b; there is a mismatch between the annotated p values in the figure legend and the annotated p values in the figure file that should be corrected.

- Please note that the white arrows are not defined in the legend of figure 1a. This needs to be rectified.

- Please add to each legend (main, EV and Appendix figures, where applicable) a 'Data Information' section explaining the statistics used or providing information regarding replicates and scales. See:

- Please add scale bars of similar style and thickness to all microscopic images (also those in the Appendix), using clearly visible black or white bars (depending on the background). Please place these in the lower right corner of the images themselves. Please do not write on or near the bars in the image but define the size in the respective figure legend. Presently, the scale bars are too thin and hard to see. Please improve.

- Please add a paragraph titled 'Biosafety' to the methods section gathering all information on where and how biosafety-relevant experiments with viruses were performed and that these were approved, and by whom (institution, government).

- Please make sure that all the funding information is also entered into the online submission system and that it is complete and similar to the one in the acknowledgement section of the manuscript text file. Presently, grants relate to the National Major Project for Infectious Disease Control and Prevention (2018ZX10731-101-001-016), the Department of Science and Technology of Jilin Province (No. 20210101015JC), the Open Project of Key Laboratory of Organ Regeneration and Transplantation, Ministry of Education, the Program for JLU Science and Technology Innovative Research Team (2017TD-08), and Fundamental Research Funds for the Central Universities are missing in the submission system. Please check.

- Please accept the tracked changes in the reagents and tools table.

In addition, I would need from you uploaded separately:

- a short, two-sentence summary of the manuscript (not more than 35 words).

- two to four short (!) bullet points highlighting the key findings of your study (two lines each).

- a schematic summary figure as separate file that provides a sketch of the major findings (not a data image) in jpeg or tiff format (with the exact width of 550 pixels and a height of not more than 400 pixels) that can be used as a visual synopsis on our website.

Best,

Referee #2:

The authors have responded to my questions and performed more tests as indicated. Alongside the work addressing the other reviewer's comments, I believe the manuscript has undergone considerable technical improvements and strengthened its conclusions.

Referee #3:

The authors have adequately addressed my previous critiques. I am happy with the changes made to the manuscript

Dear Dr. Breiling,

Thank you very much for your letter of August 8 concerning our manuscript entitled "HIV-1 Initiates Neurotoxicity Via the Vpu/Caspase 3/TDP-43 Axis" (EMBOR-2024-59135V2). We addressed each of the editor's points in the letter below and made relevant changes in the manuscript.

We are very grateful for the high quality reviews. We truly appreciate the feedback, which will substantially improve the reproducibility, quality and clarity of our paper.

Editor's comments:

- I would suggest this more comprehensive title:

HIV-1 Vpu induces neurotoxicity by promoting Caspase 3-dependent cleavage of TDP-43

Response: We sincerely appreciate the editor for this suggestion. We have made the recommended change to the title.

- Please order the manuscript sections like this, using these names:

Title page - Abstract - Keywords - Introduction - Results - Discussion - Methods - Data availability section - Acknowledgements - Disclosure and Competing Interests Statement - References - Figure legends - Expanded View Figure legends

Response: We have confirmed the order of the manuscript sections as requested.

- Please make sure that the number "n" for how many independent experiments were performed, their nature (biological versus technical replicates), the bars and error bars (e.g. SEM, SD) and the test used to calculate p-values is indicated in the respective figure legends (main, EV and Appendix figures). Please also check that all the p-values are explained in the legend, and that these fit to those shown in the figure. Please provide statistical testing where applicable. Please avoid the phrase 'independent experiment', but clearly state if these were biological or technical replicates. Please also indicate (e.g. with n.s.) if testing was performed, but the differences are not significant. In case n=2, please show the data as separate datapoints without error bars and statistics. See also:

<http://www.embopress.org/page/journal/14693178/authorguide#statisticalanalysis>

Response: As suggested by the editor, we have checked and confirmed that all the relevant information has been incorporated into the manuscript (page 21,

lines 647-648; page 22, lines 664-665 and 683-684; page 23, lines 696-697, lines 716-717; page 24, lines 728-729; page 25, lines 779-780, lines 789-790).

If $n < 5$, please show single datapoints for diagrams. It seems that presently some diagrams have no or only partial stats or the 'n.s.' is missing.

Response: We thank the editor for this suggestion, we have incorporated these items into the revised manuscript (new figures 2C-G, 4E, 5B, 5C, 5G, 6A, 6C-E, EV4B, EV4D, EV5D, S5, S6B. page 21, lines 656-657; page 23, lines 696, 703, 705, 713, 722, 726; page 24, lines 727, 728; page 25, lines 770, 774-775, 789).

Moreover:

- Please note that the legends for figures EV 5b-c is not provided in the sequential manner (legend for figure 5c is provided before legend of figure 5b).

This needs to be rectified.

Response: As suggested by the editor, the legends of Figure EV5 have been corrected as recommended (page 25, lines 784-786).

- Please note that the exact p values are not provided in the legends of figures 1d; 2h; 3f; 4e; 5e, g, i; 6a, e.

Response: The data analysis in this paper was performed using GraphPad commercial software. In this software, p-values are reported to four decimal places and rounded to " <0.0001 " when they are less than 0.0001.

- Please note that in figure EV 4b; there is a mismatch between the annotated p values in the figure legend and the annotated p values in the figure file that should be corrected.

Response: Thank you very much for pointing this out, we have corrected this item in the revised figure EV4b.

- Please note that the white arrows are not defined in the legend of figure 1a. This needs to be rectified.

Response: Thank you very much. We have included the description in the revised manuscript (page 21, lines 634).

- Please add to each legend (main, EV and Appendix figures, where applicable) a 'Data Information' section explaining the statistics used or providing information regarding replicates and scales. See:

Response: We have included a 'Data Information' section in each legend (page 21, lines 647-648; page 22, lines 664-665, and 683-684; page 23, lines 696-697, lines 716-717; page 24, lines 728-729; page 25, lines 779-780 and 789-790).

- Please add scale bars of similar style and thickness to all microscopic images (also those in the Appendix), using clearly visible black or white bars (depending on the background). Please place these in the lower right corner of the images themselves. Please do not write on or near the bars in the image but define the size in the respective figure legend. Presently, the scale bars are too thin and hard to see. Please improve.

Response: We have improved the scale bars in the revised figures 1A, 2I, 3G, 5A, 5D, 6B, 6F, EV1C, EV1D, EV3A, S3A, S4C, as recommended.

- Please add a paragraph titled 'Biosafety' to the methods section gathering all information on where and how biosafety-relevant experiments with viruses were performed and that these were approved, and by whom (institution, government).

Response: As suggested by the editor, we have incorporated these items into the revised manuscript (page 16, lines 436-439).

- Please make sure that all the funding information is also entered into the online submission system and that it is complete and similar to the one in the acknowledgement section of the manuscript text file. Presently, grants relate to the National Major Project for Infectious Disease Control and Prevention (2018ZX10731-101-001-016), the Department of Science and Technology of Jilin Province (No. 20210101015JC), the Open Project of Key Laboratory of Organ Regeneration and Transplantation, Ministry of Education, the Program for JLU Science and Technology Innovative Research Team (2017TD-08), and Fundamental Research Funds for the Central Universities are missing in the submission system. Please check.

Response: Thank you very much for pointing this out. We have entered this information into the online submission system.

- Please accept the tracked changes in the reagents and tools table.

Response: We have corrected this item in the revised version.

In addition, I would need from you uploaded separately:

- a short, two-sentence summary of the manuscript (not more than 35 words).

- two to four short (!) bullet points highlighting the key findings of your study (two lines each).

- a schematic summary figure as separate file that provides a sketch of the major findings (not a data image) in jpeg or tiff format (with the exact width of 550 pixels and a height of not more than 400 pixels) that can be used as a visual synopsis on our website.

Response: The required files have been uploaded separately as recommended by the editor.

Prof. Wei Wei
Jilin University
First Hospital
China

Dear Prof. Wei,

I am very pleased to accept your manuscript for publication in the next available issue of EMBO reports. Thank you for your contribution to our journal.

Yours sincerely,
